# Neural Likelihood Approximation for Integer Valued Time Series Data

**Luke O'Loughlin**  *luke.oloughlin@adelaide.edu.au*
*Department of Mathemaical Sciences*
*University of Adelaide*

**John Maclean**  *john.maclean@adelaide.edu.au*
*Department of Mathemaical Sciences*
*University of Adelaide*

**Andrew Black**  *andrew.black@adelaide.edu.au*
*Department of Mathemaical Sciences*
*University of Adelaide*

**Reviewed on OpenReview:** *https://openreview.net/forum?id=MMjRBe4oKF*

## Abstract

Stochastic processes defined on integer valued state spaces are popular within the physical and biological sciences. These models are necessary for capturing the dynamics of small systems where the individual nature of the populations cannot be ignored and stochastic effects are important. The inference of the parameters of such models, from time series data, is challenging due to intractability of the likelihood. To work at all, current simulation based inference methods require the generation of realisations of the model *conditional* on the data, which can be both tricky to implement and computationally expensive. In this paper we instead construct a neural likelihood approximation that can be trained using *unconditional* simulation of the underlying model, which is much simpler. We demonstrate our method by performing inference on a number of ecological and epidemiological models, showing that we can accurately approximate the true posterior while achieving significant computational speed ups compared to current best methods.

## 1 Introduction

Mechanistic models where the state consists of integer values are ubiquitous across all scientific domains, but particularly in the physical and biological sciences (van Kampen, 1992; Wilkinson, 2018). Integer valued states are important when the population being modelled is small and so the individual nature of the population cannot be ignored. Models of these types are used extensively for modelling in epidemiology (Allen, 2017), ecology (McKane & Newman, 2005), chemical reactions (Wilkinson, 2018), queuing networks (Breuer & Baum, 2005) and gene regulatory networks (Shea & Ackers, 1985), to name but a few examples.

Typically the main interest is in learning or inferring the model parameters, rather than unobserved latent states, as the parameters are directly related to the individual level mechanisms that drive the overall dynamics. In spite of this, most methods for actually performing such inference typically still involve generating realisations of the full hidden process, commonly known as simulation based inference (Cranmer et al., 2020). Particle marginal Metropolis Hastings (Doucet et al., 2015; Chopin & Papaspiliopoulos, 2020) and approximate Bayesian computation (Sisson et al., 2019) are two well known methods in this class.

As with most state-space models, the underlying state is only partially observed, but a particular feature of many integer valued models is that the subset of the state that is observed is done so accurately. The typical situation includes highly informative observations of parts of the state, while other parts are completely

unobserved Del Moral et al. (2015). Unconditional simulation of the model is unable to generate realisations of the process that are 'close' to the data, which are effectively rare events and hence generated with low probability (Drovandi & McCutchan, 2016). Thus the application of simulation based inference methods for this class of models requires simulation of realisations conditional on the data to work (Bickel et al., 2008). Typically this involves the use of importance sampling or control variates to guide simulations (Lin et al., 2013; Black, 2019). These strategies can work well, but there is a fundamental limitation that better realisations (i.e. those closer to the data) are more computationally expensive to generate and hence this often offsets any gains in the overall running time. Importance sampling can also be difficult to implement correctly and has to be done on a model by model basis—it is the opposite of a black box algorithm (Owen, 2013). In this paper we take a different approach and instead construct a neural conditional density estimator (NCDE) for modelling the full likelihood of integer valued time series data that can be trained using *unconditional* simulations of the model. Such simulations are both easy to implement and much faster to run.

The NDCE likelihood model is designed to be autoregressive using a convolutional neural network (CNN) architecture with an explicit casual structure to model conditional likelihoods by a mixture of discretised logistic distributions. The likelihood approximation is trained by modifying the sequential neural likelihood algorithm (Papamakarios et al., 2019) to jointly train the model and sample from an approximation of the posterior. We demonstrate our methods by performing inference on simulated data from two epidemic models, as well as a predator-prey model. The resulting posteriors are compared to those obtained using the exact sampling method particle marginal Metropolis Hastings (PMMH) (Andrieu et al., 2010), which is currently the state-of-the-art method (Black, 2019). We are able to demonstrate accuracy in the posteriors generated from our method, suggesting that the autoregressive model is able to learn a good approximation of the true likelihood. Furthermore, our method is significantly less computationally expensive to run in scenarios where PMMH struggles.

## 2 Problem Setup

The models we consider are continuous-time Markov chains (CTMC), $\{\boldsymbol{X}(t), t \geq 0\}$, also known as a Markov jump process (Ross, 2000; Wilkinson, 2018). The state of the system, $\boldsymbol{X}(t)$, is a vector of positive integers, which typically represent populations sizes or counts of events. The dynamics of such models are specified by describing the possible transitions in the state and the rates at which these occur. The possible state transitions are fixed for a given model, but the rates depend on small number of parameters, $\boldsymbol{\theta}$, some or all of which we wish to infer from partial time-series data, $\boldsymbol{y}_{1:n}$, generated from the model. Our observation model is that parts of the state are observed exactly, but other parts not at all. Formally the observation likelihood can be written as a product of delta functions

$$p(\boldsymbol{y}|\boldsymbol{x}) = \prod_{j \in B} \delta_{\{x^j = y^j\}} \tag{1}$$

where $x^j$ represents the $j$th component of the state and $B$ is the set of components that are assumed observed.

Although the models evolve in continuous time, we assume that observations are made at discrete time intervals and only depend on the state of the system at that time, so the inference problem fits within the a state-space model (SSM) framework (Del Moral, 2004; Murphy, 2023). Denoting the latent state and the observed data at time step $i$ as $\boldsymbol{x}_i$ and $\boldsymbol{y}_i$ respectively, then the joint likelihood is given by

$$p(\boldsymbol{x}_{1:n}, \boldsymbol{y}_{1:n}|\boldsymbol{\theta}, \boldsymbol{x}_0) = \prod_{i=1}^{n} p(\boldsymbol{y}_i|\boldsymbol{x}_i)p(\boldsymbol{x}_i|\boldsymbol{x}_{i-1}, \boldsymbol{\theta}), \tag{2}$$

where $p(\boldsymbol{x}_i|\boldsymbol{x}_{i-1}, \boldsymbol{\theta})$ is the Markov transition kernel of the model and observations are assumed conditionally independent of all previous states/observations given the current state (Del Moral, 2004).

In the Bayesian setting, we wish to sample from the posterior distribution of $\boldsymbol{\theta}$ given by

$$p(\boldsymbol{\theta}|\boldsymbol{y}_{1:n}) \propto p(\boldsymbol{y}_{1:n}|\boldsymbol{\theta})p(\boldsymbol{\theta}),$$

where $p(\boldsymbol{\theta})$ is the prior and $p(\boldsymbol{y}_{1:n}|\boldsymbol{\theta})$ the likelihood. Typically this is done using Markov chain Monte Carlo (MCMC), but this is difficult for this class of models as the likelihood is intractable. This is because, although in theory the transition kernel $p(\boldsymbol{x}_i|\boldsymbol{x}_{i-1}, \boldsymbol{\theta})$ can be numerically computed as a matrix exponential, in practice this is intractable for all but the simplest or smallest systems (Black et al., 2017; Sherlock, 2021). Hence the likelihood, obtained by marginalising over $\boldsymbol{x}_{1:n}$ in Equation 2, is also intractable.

While the likelihood itself is intractable, simulation of CTMCs (sampling from the transition density $p(\boldsymbol{x}_i|\boldsymbol{x}_{i-1}, \boldsymbol{\theta})$) is straightforward (Gillespie, 1976). Particle filter approaches are a natural and popular way of estimating the likelihood, which can then be used for inference of the parameters using particle marginal Metpopolis-Hastings (Wilkinson, 2018; Chopin & Papaspiliopoulos, 2020). For the class of problems we are interested in, with perfect but incomplete observations, the difficulty implementing particle filters stems from the form of the observation likelihood in Equation 1. As this equation is the product of delta functions, only realisations of the latent process that exactly match the observations are assigned non-zero likelihood. If the state space is naturally large, or the proposed parameters are not compatible with the observations, then producing simulations that match the data becomes a rare event problem. Conditional simulation methods (i.e. simulating realisations of the model conditional on the observed data using importance sampling) can be used to guide the simulations to match the data (Black, 2019). We use these as a gold standard comparison in this paper, but as we also show they can become computationally expensive in certain situations. Our approach, detailed in the next section, is to instead construct a NCDE that models the likelihood $p(\boldsymbol{y}_{1:n}|\boldsymbol{\theta})$ directly. This still relies on simulations of the model for training, but these are basic unconditional simulations that are computationally cheap.

## 3 Methods

In this section, we construct an autoregressive model to approximate $p(\boldsymbol{y}_{1:n}|\boldsymbol{\theta})$. The simplest version of the model works with univariate time series data $y_{1:n}$ and is described in Section 3.1. We break up the construction into two parts: First, in Section 3.1.1 we describe a CNN architecture using causal convolutions (van den Oord et al., 2016), which is used to map the input sequence $y_{1:n}$ to an output sequence $\boldsymbol{o}_{1:n}$ such that $\boldsymbol{o}_i$ "does not look into the future" (it is only dependent on $y_{1:i}$ and $\boldsymbol{\theta}$). Second, in Section 3.1.2 we describe a way to map each $\boldsymbol{o}_i$ onto a probability distribution supported on the set of integers $\mathbb{Z}$ (or a subset thereof), so that we can use $\boldsymbol{o}_i$ to approximate $p(y_{i+1}|y_{1:i}, \boldsymbol{\theta})$. In Section 3.2, we describe how the autoregressive model can be extended to multidimensional integer valued time series $\boldsymbol{y}_{1:n}$. In particular, this extension allows the autoregressive model to model correlations between the features of $\boldsymbol{y}_i$. Finally, in Section 3.3 we provide an overview of sequential neural likelihood (SNL), which is an algorithm for simultaneously training the surrogate likelihood and performing approximate inference (Papamakarios et al., 2019). We also discussing some modifications to SNL to help increase consistency in the approximate posterior.

### 3.1 Autoregressive Model

Autoregressive models approximate the density of a time series $\boldsymbol{y}_{1:n}$ by modelling a sequence of conditionals, namely $p(\boldsymbol{y}_i|\boldsymbol{y}_{1:i-1})$, exploiting causality in the time series (Shumway & Stoffer, 2017). We will denote the autoregressive model conditionals by $q_\psi(\boldsymbol{y}_i|\boldsymbol{y}_{1:i-1})$, where $\psi$ are the autoregressive model parameters[1], e.g. neural network weights. Suppressing the dependence on $\boldsymbol{\theta}$, the joint density for the likelihood can be factorised as

$$p(\boldsymbol{y}_{1:n}) = p(\boldsymbol{y}_1) \prod_{i=2}^{n} p(\boldsymbol{y}_i|\boldsymbol{y}_{1:i-1}),$$

then we can define the density of $\boldsymbol{y}_{1:n}$ under the autoregressive model in the same way as

$$q_\psi(\boldsymbol{y}_{1:n}) \equiv q_\psi(\boldsymbol{y}_1) \prod_{i=2}^{n} q_\psi(\boldsymbol{y}_i|\boldsymbol{y}_{1:i-1}). \tag{3}$$

---

[1]Note that these parameters are unrelated to the parameters of the CTMC $\boldsymbol{\theta}$.

The parameters $\psi$ are learnt from training data, using Equation 3 to construct a suitable loss function. The standard choice for the loss is the negative log-likelihood (NLL) (Shumway & Stoffer, 2017), and it is what we use here. In particular, for a dataset consisting of $N$ time series $\{\boldsymbol{y}_{1:n}^{(j)}\}_{j=1}^{N}$, the NLL loss leads to minimisation of an unbiased estimate of the forward KL-divergence $\hat{D}_{KL}(p||q_\psi)$:

$$\hat{D}(p||q_\psi) = \frac{1}{N}\sum_{j=1}^{N}\log\frac{p\left(\boldsymbol{y}_{1:n}^{(j)}\right)}{q_\psi\left(\boldsymbol{y}_{1:n}^{(j)}\right)},$$

$$= -\frac{1}{N}\sum_{j=1}^{N}\log q_\psi\left(\boldsymbol{y}_{1:n}^{(j)}\right) + C,$$

where $C$ is a constant with respect to $\psi$. Since the KL divergence is non-negative and equal to 0 if and only if $q_\psi(\boldsymbol{y}_{1:n}) = p(\boldsymbol{y}_{1:n})$ (Bishop, 2006), it follows that a sufficiently flexible autoregressive model will be able to approximate the true density arbitrarily well in the large dataset limit. When conditioning on $\boldsymbol{\theta}$, the NLL loss is still appropriate, as if $\boldsymbol{\theta} \sim p(\boldsymbol{\theta})$ for some proposal $p(\boldsymbol{\theta})$, then as pointed out by Papamakarios et al. (2019), in the large $N$ limit

$$\frac{1}{N}\sum_{j=1}^{N}\log\frac{p\left(\boldsymbol{y}_{1:n}^{(j)}|\boldsymbol{\theta}^{(j)}\right)}{q_\psi\left(\boldsymbol{y}_{1:n}^{(j)}|\boldsymbol{\theta}^{(j)}\right)} \longrightarrow \mathbb{E}_{\boldsymbol{\theta}\sim p(\boldsymbol{\theta})}\left[D_{KL}(p(\boldsymbol{y}_{1:n}|\boldsymbol{\theta})||q_\psi(\boldsymbol{y}_{1:n}|\boldsymbol{\theta}))\right].$$

The right hand side is non-negative and equal to zero if and only if $p(\boldsymbol{y}_{1:n}|\boldsymbol{\theta}) = q_\psi(\boldsymbol{y}_{1:n}|\boldsymbol{\theta})$ almost everywhere on the support of $p(\boldsymbol{\theta})$.

Since we are interested in using $q_\psi(\boldsymbol{y}_{1:n}|\boldsymbol{\theta})$ as a surrogate within an MCMC scheme, where it is repeatedly evaluated, it is preferable to use a model architecture which allows for efficient evaluation of Equation 3. For univariate time series, CNN based architectures have been proposed which allow for parallel evaluation of all of the terms in Equation 3. These CNNs, called causal CNNs, map the univariate input sequence $y_{1:n}$ to the sequence of log-conditionals $(\log q_\psi(y_i|y_{1:i-1}))_{1:n}$. A typical example is the model WaveNet (van den Oord et al., 2016), which uses a causal CNN[2] to synthesise audio data. Due to the efficiency of convolutions in deep learning, we construct our autoregressive model using causal convolutions. Additionally, automatic differentiation allows for computation of the gradient of the causal CNN surrogate (with respect to the model parameters $\boldsymbol{\theta}$), and this has the same complexity as the evaluation (Baydin et al., 2017), so efficient MCMC samplers such as NUTS (Hoffman & Gelman, 2014) can be employed to sample from the posterior rather than basic random walk Metropolis Hastings samplers, which are much less efficient.

### 3.1.1 Causal Convolutions

A one-dimensional causal convolution can be understood as a regular one-dimensional convolution with appropriate zero padding on the left of the input. For a convolution kernel with length $k$ and $d_c$ output channels, say $\boldsymbol{w}_{1:k} \in \mathbb{R}^{d_c \times k}$, then the causal convolution operation is defined by

$$c_k(\boldsymbol{w}_{1:k}, y_{1:n}) = \boldsymbol{w}_{1:k} * (\overbrace{0, \cdots, 0}^{k-1}, y_{1:n})$$

$$\implies c_k(\boldsymbol{w}_{1:k}, y_{1:n})_i = \sum_{j=0}^{k-1}\boldsymbol{w}_{k-j}y_{i-j}\mathbf{1}_{j<i},$$

where we have used $*$ to denote the standard convolution operation. It is clear that $c_k(\boldsymbol{w}_{k-j}, y_{1:n})_i$ depends only on the input sequence only through the elements $y_{l(i):i}$, where $l(i) = \max\{1, i - (k-1)\}$, hence the causal structure in the input sequence is preserved by $c_k$. By constructing CNN layers using the operation $c_k$ (assuming that the kernel length is $k$ in every layer), then it is clear that composing these layers will yield a function which also preserves the causal structure of the input sequence. Moreover, denoting the output

---

[2]WaveNet also uses dilated convolutions to increase the length of the receptive field of the model.

sequence by $\boldsymbol{o}_{1:n}$, then it is straightforward to show that $\boldsymbol{o}_i$ only depends on $y_{1:n}$ through the elements $y_{l_h(i):i}$ (see Figure 1), where $h$ is the number of CNN layers and $l_h(i) = \max\{1, i - h(k+1)\}$. The purpose of constructing such a CNN is that since $\boldsymbol{o}_{i-1}$ only depends on elements up to time step $i - 1$, then it can be used to define $q_\psi(y_i | y_{1:i-1})$. To do this, $\boldsymbol{o}_{i-1}$ is used to compute the parameters of a suitable parametric distribution with tractable probability mass function, which is discussed in Section 3.1.2. To aid with training of the model, we also use residual connections in the hidden layers of the CNN, which are known to help stabilise training (He et al., 2016).

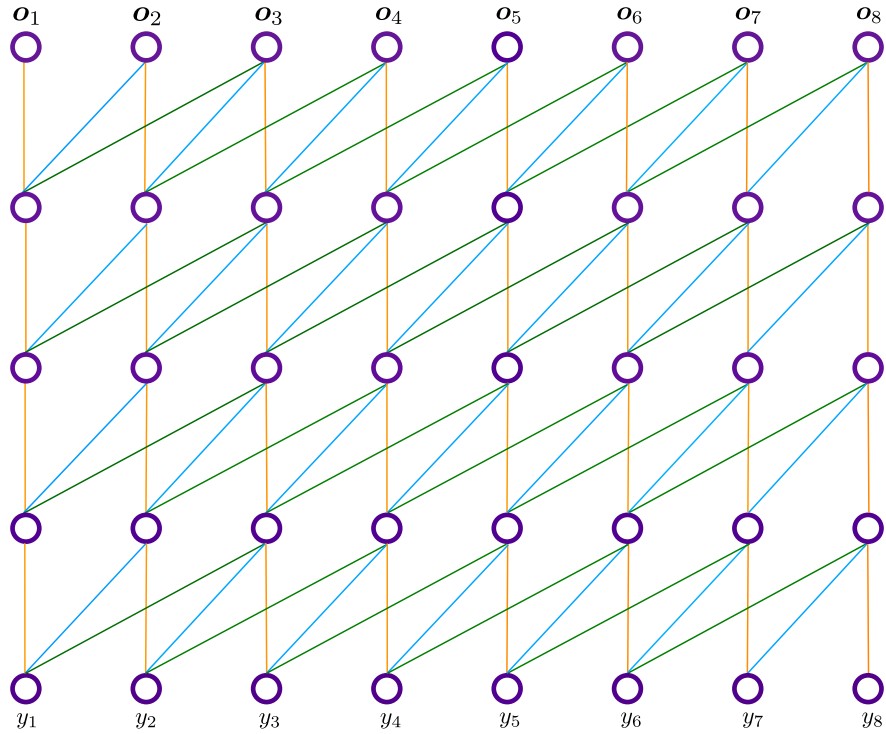

Figure 1: An example of the dependency structure in a causal convolutional neural network with 4 layers and using a convolution kernel length of 3 in every layer. Notice that paths from $\boldsymbol{o}_i$ back to the input elements $y_j$ only exist for $j \leq i$.

By construction, the causal CNN can only model conditionals of the form $p(y_i | y_{l_h(i):i-1})$ due to the finite receptive field of the neural network. In particular, the receptive field grows linearly with the convolution kernel length $k$ and the number of hidden layers in the network $h$. For the problems considered in this paper, there are no relevant long range correlations in the data, so the finite receptive field approximation is reasonable. If long range correlations between observations were important for a particular problem, then the model could be altered by dilating the convolutions as this modification produces a receptive field that grows exponentially with the number of hidden layers (van den Oord et al., 2016).

To model the likelihood $p(y_{1:n} | \boldsymbol{\theta})$, we must augment the CNN so that $\boldsymbol{o}_{i-1}$ is dependent on $\boldsymbol{\theta}$ as well as $y_{1:i-1}$. We perform the augmentation by using a shallow neural network to calculate a context vector $\mathbf{c}(\boldsymbol{\theta})$, and then adding a linear transformation of $\mathbf{c}(\boldsymbol{\theta})$ to the output of each layer of the CNN (broadcasting across the time axis). For inference problems where some parameters are unidentifiable, or where their effect on the dynamics are minimal, the neural network $\mathbf{c}$ can help to learn a suitable reparameterisation of $\boldsymbol{\theta}$, which should allow for greater accuracy in the autoregressive model. We chose to make every layer of the CNN dependent on $\boldsymbol{\theta}$, as this allows for greater interaction between the model parameters and observations. This is important for non-linear models where small changes in the model parameters can cause noticeable, or even substantial changes in the observation dynamics (Kuznetsov, 2004; Börgers, 2017). Figure 2 shows a diagram of the details of the evaluation of the CNN.

Further details of the model architecture are given in Appendix A of the supplement.

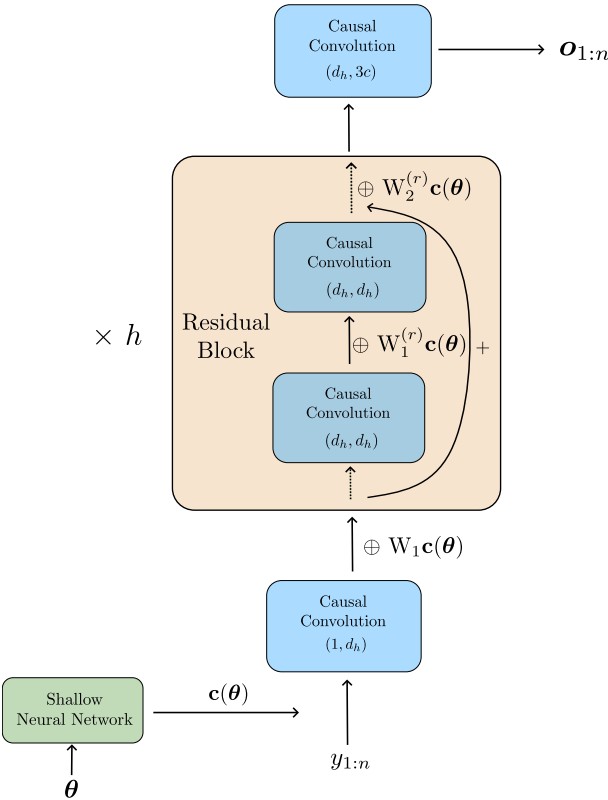

Figure 2: Evaluation of the causal CNN for one-dimensional inputs. The numbers $(a, b)$ in the causal convolution blocks denote the number of input/output channels for the convolution layer. The quantity $d_h$ denotes the number of channels used in the hidden layers, and $h$ is the number of residual blocks used. We have used $\oplus$ to denote addition of a vector and sequence of vectors broadcasted across the time axis.

### 3.1.2 Discretised Mixture of Logistics

To calculate the autoregressive model conditionals from $\boldsymbol{o}_{1:n}$, we use the components of each $\boldsymbol{o}_i$ to compute the parameters of a discretised mixture of logistic distributions (DMoL) (Salimans et al., 2017). The discretised logistic (DL) is family of distributions supported on $\mathbb{Z}$, characterised by a shift parameter $\mu \in \mathbb{R}$ and a scale parameter $s > 0$. A $\mathrm{DL}(\mu, s)$ random variable $Y$ is sampled by first sampling $X \sim \mathrm{Logistic}(\mu, s)$, and then calculating $Y = \lfloor X \rfloor$, where $\lfloor \cdot \rfloor$ denotes rounding down to the nearest integer. The logistic distribution has similar properties to the Gaussian distribution, however it has an analytically expressible cumulative distribution function (CDF), which makes it possible to calculate the probability mass function

$$\Pr\left(Y = m\right) = \Pr\left(X \leq m + 1\right) - \Pr\left(X \leq m\right).$$

Denoting the probability mass function (PMF) of a discretised logistic with shift $\mu$ and scale $s$ by $p_{\mathrm{DL}}(y; \mu, s)$, then a mixture of $c$ DLs has PMF

$$p_{\mathrm{DMoL}}(y) = \sum_{i=1}^{c} \phi_i \, p_{\mathrm{DL}}(y; \mu_i, s_i), \tag{4}$$

where the $\phi_i$ terms are weights satisfying $\phi_i \geq 1$ and $\sum_{i=1}^{c} \phi_i = 1$. The DMoL serves as an analytically tractable family of distributions that is flexible enough to approximate a wide class of distributions supported on $\mathbb{Z}$, analogous to a mixture of Gaussians in the continuous case (Salimans et al., 2017). It can also helpful to truncate the discretised logistics when a problem involves data supported on a subset $\{a \leq n \leq b \mid n \in \mathbb{Z}\}$. This is easy, as it only involves renormalising the PMF of each DL component in Equation 4, which can be done analytically, noting that the DL CDF is the same as the logistic CDF restricted to $\mathbb{Z}$.

To use a DMoL to represent the autoregressive model conditionals, we use the outputs of the CNN, $\boldsymbol{o}_{1:n}$, to calculate the shift, scale and mixture proportion parameters. Assuming that $q_\psi(y_i|y_{1:i-1}, \boldsymbol{\theta})$ has $c$ mixture components, then we must calculate $3c - 1$ parameters, $c$ shifts $\{\mu_i^j\}_{j=1,\ldots,c}$, $c$ scales $\{s_i^j\}_{j=1,\ldots,c}$, and $c - 1$ mixture proportions $\{\phi_i^j\}_{j=1,\ldots,c-1}$ ($\phi_i^c$ is redundant since $\sum_j \phi_i^j = 1$). We use a CNN with $3c$ output channels (i.e. $\boldsymbol{o}_i \in \mathbb{R}^{3c}$), and then the parameters of the DMoL are given by

$$\mu_i^j \equiv y_{i-1} + o_i^j, \tag{5a}$$

$$s_i^j \equiv \text{softplus}(o_i^{c+j}) + \epsilon, \tag{5b}$$

$$(\phi_i^1, \cdots, \phi_i^c) \equiv \text{softmax}\left((o_i^{2c+1}, \cdots, o_i^{3c})\right), \tag{5c}$$

where $o_i^j$ is the $j^{\text{th}}$ element of $\boldsymbol{o}_i$ and $\epsilon$ is a small constant for numerical stability which we take to be $10^{-6}$. Note that Equations 5(a-c) ensures that $s_i^j > \epsilon$ and $\phi_i^j$ for all $i$, $j$, and $\sum_j \phi_i^j = 1$ for all $i$, so the conditionals $q_\psi(y_i|y_{1:i-1}, \boldsymbol{\theta})$ are well defined distributions.

For some models we alter the default expressions in Equation 5 to better reflect the particular structure of a model. For example, in Sections 4.1 and 4.2 the epidemic models we investigate produce count data which is bounded above by the total population size $N$, hence we scale $s_i^j$ by $(N - y_{i-1})/N$ to reflect the shrinking support of the conditionals over time.

## 3.2 Multivariate Data

When the time series consists of multivariate observations $\boldsymbol{y}_i$, then constructing the autoregressive model is less straightforward. The main hurdle is to replace the DMoL distribution used in the univariate case with a suitable parametric family of multivariate discrete distributions. For $\boldsymbol{y}_i \in \mathbb{Z}^d$, $d > 1$, the conditionals $p(\boldsymbol{y}_i|\boldsymbol{y}_{1:i-1}, \boldsymbol{\theta})$ can be factorised as follows:

$$p(\boldsymbol{y}_i|\boldsymbol{y}_{1:i-1}, \boldsymbol{\theta}) = p(y_i^1|\boldsymbol{y}_{1:i-1}, \boldsymbol{\theta}) \prod_{j=2}^d p(y_i^j|\boldsymbol{y}_i^{1:j-1}, \boldsymbol{y}_{1:i-1}, \boldsymbol{\theta}), \tag{6}$$

where $y_i^j$ is the $j^{\text{th}}$ component of $\boldsymbol{y}_i$ and $\boldsymbol{y}_i^{1:j-1}$ is a vector of the first $j - 1$ components of $\boldsymbol{y}_i$. It follows that the problem of modelling $p(\boldsymbol{y}_i|\boldsymbol{y}_{1:i-1}, \boldsymbol{\theta})$ can be approached by modelling the $d$ univariate conditionals on the right hand side of Equation 6. We will denote the univariate conditionals approximated by the autoregressive model by $q_\psi(y_i^j|\boldsymbol{y}_i^{1:j-1}, \boldsymbol{y}_{1:i-1}, \boldsymbol{\theta})$, and define the full autoregressive model conditionals by

$$q_\psi(\boldsymbol{y}_i|\boldsymbol{y}_{1:i-1}, \boldsymbol{\theta}) \equiv q_\psi(y_i^1|\boldsymbol{y}_{1:i-1}, \boldsymbol{\theta}) \prod_{j=2}^d q_\psi(y_i^j|\boldsymbol{y}_i^{1:j-1}, \boldsymbol{y}_{1:i-1}, \boldsymbol{\theta}).$$

The CNN architecture described in Section 3.1.1 is not immediately suitable for calculation of $q_\psi(y_i^j|\boldsymbol{y}_i^{1:j-1}, \boldsymbol{y}_{1:i-1}, \boldsymbol{\theta})$. This is because to construct a suitable model for $q_\psi(y_i^j|\boldsymbol{y}_i^{1:j-1}, \boldsymbol{y}_{1:i-1}, \boldsymbol{\theta})$, the $i^{\text{th}}$ output element of the CNN $\boldsymbol{o}_i$ must decompose in a way such that there is a group of elements which only depend on $\boldsymbol{y}_i$ through $\boldsymbol{y}_i^{1:j-1}$, which can then be used to define $q_\psi(y_i^j|\boldsymbol{y}_i^{1:j-1}, \boldsymbol{y}_{1:i-1}, \boldsymbol{\theta})$. However using the CNN from Section 3.1.1 yields an output $\boldsymbol{o}_i$ that is dependent on $\boldsymbol{y}_i$ in an arbitrary way, hence $\boldsymbol{o}_i$ cannot be split like this. A simple solution to this issue is to notice that we may 'unroll' the time series $\boldsymbol{y}_{1:n}$ into a long univariate time series

$$z_{1:dn} = \left(y_1^1, y_1^2, \ldots, y_1^d, y_2^1, \ldots, y_n^{d-1}, y_n^d\right),$$

and then the CNN architecture from Section 3.1.1 can be used on $z_{1:dn}$ to define

$$q_\psi(z_{id+j}|z_{1:id+j-1}, \boldsymbol{\theta}) = q_\psi(y_i^j|\boldsymbol{y}_i^{1:j-1}, \boldsymbol{y}_{1:i-1}, \boldsymbol{\theta}).$$

In practice, we do not unroll the time series, and instead force certain weight matrices in the convolution kernels of the CNN to have a block lower triangular structure. The block lower triangular structure of the weight matrices is achieved with binary masking as in the density model MADE (Germain et al., 2015). The

block lower triangular structure ensures that $\boldsymbol{o}_i$ can be partitioned into $j$ blocks of elements, where the first block is not dependent on $\boldsymbol{y}_i$, and the $j^{th}$ block is dependent $\boldsymbol{y}_i^{1:j-1}$ for $j > 1$. A more detailed discussion of the masking procedure is given in Appendix A.2. For the generic case, we use $3cd$ output channels in the CNN, so that each block (of size $3c$) of $\boldsymbol{o}_i$ can be used to calculate the $3c - 1$ parameters of a DMoL distribution, as described in Section 3.1.2. This can be altered if necessary, e.g. if one of the features is binary, then number of output channels and masks for the weight matrices can be adjusted so that only one parameter is output for the binary random variable (corresponding to the parameter of a Bernoulli distribution).

Note that by modelling the conditionals as described above, we must choose an ordering for the components of $\boldsymbol{y}_i$[3]. In general, such an ordering will not be natural, and although in theory the decomposition in Equation 6 will be agnostic to the choice of ordering, in practice the neural network will be able to represent certain orderings more easily than others. Hence, the training and inference may be more stable for some orderings of the components compared to others, and problem specific knowledge may be required to find a sensible choice for the ordering.

### 3.3 Sequential Neural Likelihood

Sequential neural likelihood (SNL) is a SBI algorithm which uses a NCDE as a surrogate for the true likelihood within a MCMC scheme (Papamakarios et al., 2019). The main issue that is addressed by SNL is how to choose the proposal distribution over parameters in the training dataset $\tilde{p}(\boldsymbol{\theta})$. Ideally, the proposal would be the posterior over the model parameters so that the surrogate likelihood is well trained in regions where the MCMC sampler is likely to explore, without wasting time learning the likelihood in other regions. Obviously the posterior is not available, so instead SNL iteratively refines $q_\psi$ by sampling additional training data over a series of rounds. The initial proposal is taken to be the prior, and subsequent proposals are taken to be an approximation of the posterior, which is sampled from using MCMC with the surrogate from the previous round as an approximation of the likelihood. We refer the reader to Papamakarios et al. (2019) for more details on SNL.

We use our autoregressive model $q_\psi(\boldsymbol{y}_{1:n}|\boldsymbol{\theta})$ within SNL to perform inference for the experiments described in Section 4. We make a modification to how we represent the SNL posterior, in that we run SNL over $r$ rounds, and keep a mixture of MCMC samples over the final $r' < r$ rounds. The resulting approximate posterior is therefore an equally weighted mixture of approximations over successive SNL rounds. We do this because sometimes the posterior statistics exhibit variability which does not settle down over the SNL rounds, so it can be difficult to assess whether the final round is truly the most accurate. Instead we use an ensemble to average out errors present in any given round, which worked well in practice. Variability across the SNL rounds occurs mostly when the length of time series is random, presumably because the random lengths inflate variability in the training dataset.

## 4 Experiments

We evaluate our methods on simulated data from three different models, comparing the results to the exact sampling method PMMH (Andrieu et al., 2010). The first two models we consider are epidemic models, namely the SIR and SEIAR model, the former being a simple but ubiquitous model in epidemiology (Allen, 2017), and the latter being a more complex model which admits symptomatic phases (Black, 2019). The third example is the predator-prey model described in McKane & Newman (2005).

We perform two sets of experiments using the SIR model; one set considers a single simulated outbreak in a population of size 50, and the other considers multiple independent observations from outbreaks within households (small closed populations). The experiments on the single outbreak serve as a proof of concept for our method, as PMMH can target the posterior for this data efficiently (Black, 2019). The experiments on household data allow us to assess the scalability of our method with an increasing number of independent time series, where PMMH does not scale well due to the exponential increase in the number of particles needed to keep the variance of the likelihood estimate low enough (Sherlock et al., 2015).

---

[3]The ordering must be fixed, i.e. it cannot depend on $i$.

The SEIAR model experiments serve two purposes, one being to assess the accuracy of our method when trained on data from a more complex model whose likelihood includes complex relationships between the model parameters. The other purpose is to assess the scalability with the population size, as done in Black (2019), as increasing population sizes generally causes the performance of PMMH to degrade due to combination of longer time series and a larger model state space. The predator-prey model experiments also serve two purposes, one being to assess our method on a model with multivariate observations, and the other being to assess the scalability with lowering observation errors, where PMMH again generally performs poorly due to weight degeneracy (Del Moral & Murray, 2015).

We run SNL 10 times with a different random seed for every outbreak experiment, and 5 times for the predator prey model experiments. Each run of SNL uses 10 rounds with NUTS (Hoffman & Gelman, 2014) as the MCMC sampler, and using the final 5 rounds to represent the posterior. Unless otherwise stated, we used a simulation budget of 60,000 samples for SNL, beginning with 10,000 samples from the prior predictive. All neural network models were constructed using JAX (Bradbury et al., 2018) and the MCMC was run using numpyro (Phan et al., 2019). We validate our method against PMMH: for all outbreak experiments we use the particle filter of Black (2019) as it is specifically designed for noise free outbreak data. For the predator-prey model, we use a bootstrap filter (Kitagawa, 1996). We tuned the parameters of PMMH (e.g. number of particles, the Metropolis-Hastings proposal) based on a pilot run, and subsequently ran the tuned PMMH chain for as many MCMC iterations as necessary to generated a target effective sample size (ESS) of at least 1,000.

To assess the accuracy of our method, we compare the posterior means and standard deviations for each parameter obtained from PMMH and SNL. We compare the estimated posterior means relative to the prior standard deviation (denoted M) to account for variability induced by using wide priors. We also compare the ratio of the estimated standard deviations, shifted so that the metric is 0 when SNL is accurate (denoted S):

$$M = \frac{\mu_{\text{SNL}} - \mu_{\text{PMMH}}}{\sigma_{\text{Prior}}}, \quad S = \frac{\sigma_{\text{SNL}}}{\sigma_{\text{PMMH}}} - 1.$$

We also report a classifier 2-sample t-test (C2ST) score (Lueckmann et al., 2021) for the SEIAR model experiments, which measures the similarity of two distributions by evaluating the average score of a classifier trained to distinguish between samples from the two distributions. We do this because the 5-dimensional posteriors have complex geometries that make mean and variance comparisons insufficient in assessing the discrepancies between the two posteriors. In the case of the predator-prey experiments, achieving a large enough ESS from the PMMH posterior to train a classifier becomes far more expensive, which is why we omit the C2ST score in this case. All posterior pairs plots can be found in Appendix D for a qualitative comparison between the results. We assess the runtime performance of our method by calculating the effective sample size (of the MCMC samples) per second (ESS/s) averaged over the parameters, where we take the ESS for SNL to be an average of the ESS values obtained from the MCMC chains over the final 5 rounds. For all experiments, the chosen simulation budget ensured that we generated an ESS of approximately 1,000-3,000 per SNL round. More detailed descriptions of the models and additional details for the experiments such as hyperparameters can be found in Appendix B.

## 4.1 SIR model

We first consider outbreak data from the SIR model (Allen, 2017). This model is a CTMC whose state at time $t$ counts the number of susceptible and infected individuals, $S_t$ and $I_t$ respectively. Equivalently, the state can be given in terms of two different counting processes: the total number of infections $Z_{1,t} = N - S_t$ and removals $Z_{2,t} = N - (S_t + I_t)$, where $N$ is the population size. Our observed data is a time series of total case numbers $y_{1:n}$, where $y_i = Z_{1,i}$, equivalent to recording the number of new cases every day without error. Additionally, we assume that the outbreak has ended by day $n$ so the final size of the outbreak is equal to $y_n$. Since the outbreak never grows beyond $y_n$ cases, we can replace the full time series by $y_{1:\tau}$, where $\tau$ is random, and defined as the smallest $i$ with $y_i = y_n$. The task is to infer the two parameters of the model, $R_0 > 0$ and $1/\gamma > 0$, using the data $y_{1:\tau}$. To model the randomness in $\tau$, we augment the autoregressive model with the binary sequence $f_{1:\tau}$, where $f_i = \delta_{i,\tau}$. The autoregressive model for this data is constructed by using

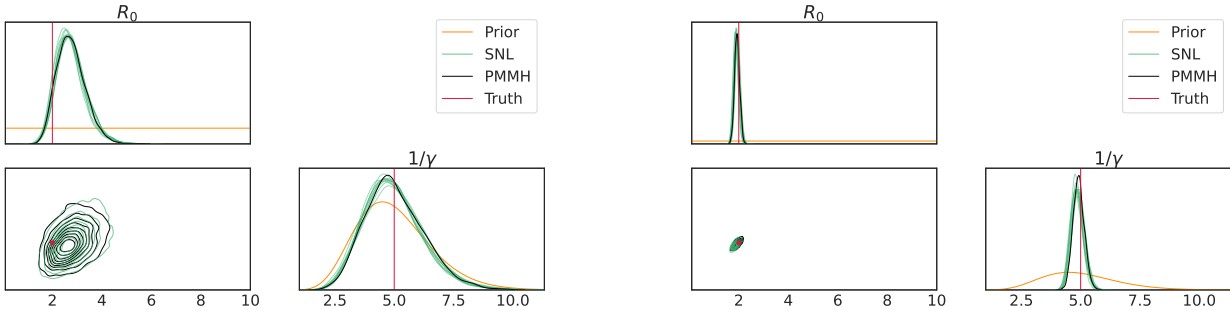

Figure 3: Posterior Pairs Plots for SIR Model Experiments. Left: single outbreak experiment from a population of $N = 50$. Right: experiment using 500 households. For the univariate posteriors, we show all 10 runs of SNL, but only one of the runs for the bivariate posterior.

$y_{1:\tau}$ as input to the CNN, and adding an additional output channel to the final layer of the CNN, which allows for calculation of $q_\psi(f_i|y_{1:i}, f_{1:i-1}, \boldsymbol{\theta})$. Using the multivariate formulation of the autoregressive model would be redundant, as by construction we always have $p(y_i, f_i|y_{1:i-1}, f_{1:i-1}, \boldsymbol{\theta}) = p(y_i, f_i|y_{1:i-1}, f_{i-1}, \boldsymbol{\theta})$, and $f_{i-1} = 0$ for any $i \leq \tau$; this fact and more details concerning the autoregressive model construction in these experiments are explained in Appendix B.2.

**Single realisation** Our proof of concept example is a single observed outbreak in a population of size 50. The left of Figure 3 shows the posterior pairs plot of SNL compared to PMMH, which clearly shows that SNL provides an accurate approximation of the true posterior for this experiment; see Appendix C for a quantitative comparison. For this particular experiment, PMMH obtains an ESS/s of 69.38, which outperforms SNL with only 3.74. This is unsurprising, as we are not using a large amount of data and the particle filter we used was specifically designed to perform well for observations of this form.

**Household data** Here, we mimic outbreak data collected from households (Walker et al., 2017), by generating $h$ observations with household sizes uniformly sampled between 2 and 7. We use values $h = 100$, 200 and 500, assume that all households are independent, and that all outbreaks are governed by the same values of $R_0$ and $1/\gamma$, which have been chosen to be typical for this problem. The likelihood of an observation where $y_\tau = 1$, or where the household size is 2, can be calculated analytically, so we incorporate these terms into inference manually. We therefore train the autoregressive model to approximate $p(y_{1:\tau}|\boldsymbol{\theta}, y_\tau > 1)$ for household sizes of 3 through 7. For all of these experiments, we increased the simulation budget to 100,000 as the observations are much shorter and less informative than in the previous set of experiments.

Table 1 shows the quantitative comparison between the two posteriors, indicating a negligible bias in the SNL means on average. The SNL variance appears to be slightly inflated on average, mainly for the $h = 500$ experiments, though this is partly due to the PMMH variance shrinking with $h$, so the difference between the PMMH and SNL posteriors is still small, as indicated by the right of Figure 3. The ESS/s of SNL and PMMH are compared in the left panel of Figure 4, which clearly shows that SNL outperforms PMMH across each experiment, and that the drop off in SNL's performance is less significant for increasing $h$. This is mainly because the autoregressive model can be evaluated efficiently on the observations in parallel.

### 4.2 SEIAR model

This more sophisticated outbreak model includes a latent exposure period (E), symptomatic phases (I) and asymptomatic (A) phases (Black, 2019). The state of the model can be given in terms of 5 counting processes $Z_{1,t}, \ldots, Z_{5,t}$, which count the total number of exposures, pre-symptomatic cases, symptomatic cases, removals and asymptomatic cases respectively. In addition to $R_0$ and $1/\gamma$, the model has three extra parameters: $1/\sigma \geq 0$, $\kappa \in (0, 1)$ and $q \in (0, 1)$. The observations are of the same form as the SIR model, where the case numbers at day $i$ are given by $y_i = Z_{3,i}$, corresponding to counting the number of new

Table 1: Comparison of SNL and PMMH posterior statistics for household data experiments. The values reported are the average over the 10 SNL runs and the maximum deviation from 0.

| Parameter | $h$ | M | | S | |
|---|---|---|---|---|---|
| | | avg | max | avg | max |
| $R_0$ | 100 | -0.003 | -0.02 | 0.01 | -0.08 |
| | 200 | -0.002 | -0.02 | 0.02 | 0.13 |
| | 500 | -0.003 | -0.02 | 0.04 | 0.11 |
| $1/\gamma$ | 100 | 0.004 | -0.10 | 0.02 | 0.10 |
| | 200 | 0.002 | -0.05 | 0.06 | 0.15 |
| | 500 | -0.03 | -0.07 | 0.11 | 0.17 |

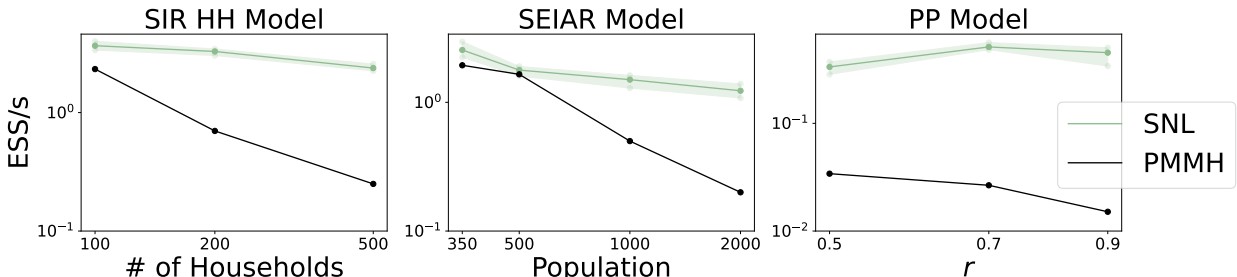

Figure 4: Comparison of ESS/s for SNL and PMMH for all three models. The lines plotted for SNL are the averages over the runs, and the error bars correspond to the maximum and minimum. All plots are on a log-log scale. The horizontal axis of the left panel scales the *observed dataset size*, showing that our method has improved scaling compared to the gold standard PMMH. The horizontal axis of the middle panel scales the *system size* and similarly shows improved scaling. The horizontal axis of the right panel scales the *observation accuracy*, as discussed in Section 4.3. We observe the well-known phenomenon that PMMH runtime degrades with more accurate data, but SNL improves.

symptomatic cases each day. The likelihood has a complicated relationship between the parameters $R_0$ and $q$, which manifests as a banana shaped joint posterior. We assess our method on observations from populations of size $N = 350,\ 500,\ 1000$ and $2000$.

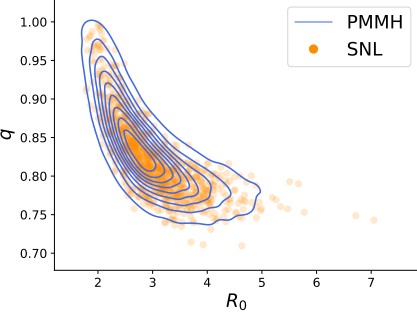

Figure 5: $(R_0, q)$ joint posterior samples for SEIAR experiment with $N = 500$.

Table 2 shows a comparison of the posterior statistics for the parameters $R_0$ and $\kappa$. The metrics for the other parameters are deferred to Appendix C. The bias in the mean value of $R_0$ is consistently small, but the standard deviations are slightly inflated, more so with larger population sizes. The $R_0$ posterior for these experiments is quite right skewed, and SNL produces posteriors which are slightly too heavy in the tail, which explains the inflated variance estimates. For $\kappa$, the posterior means do exhibit a noticeable bias, but the posterior standard deviations do not on average. The bias in $\kappa$ is lowest for the $N = 500$

experiment, where the posterior has a mode near $\kappa = 0$. The parameter $\kappa$ only has a weak relationship with the data, being the most informative near 0 or 1, hence the effect of $\kappa$ on the likelihood is presumably difficult to learn. This would explain the biases in the $\kappa$ posteriors, and would also explain why the $N = 500$ experiment had the lowest bias on average. We also report the C2ST scores in Table 3. All outputs from the classifier are very close to 0.5, which is obtained when the classifier cannot learn to distinguish between the two distributions. Additionally, a p-value can be calculated from the classifier scores under the hypothesis that the two distributions are the same (Lopez-Paz & Oquabe, 2017), and we see that all p-values are larger than 0.5 except for $N = 2,000$ (which is still larger than 0.05), suggesting that the bulk of the posterior masses must overlap significantly. For some visual confirmation of this, Figure 5 shows that SNL manages to correctly reproduce the banana shaped posterior between $R_0$ and $q$ in the $N = 500$ experiment. In terms of runtime, the centre panel of Figure 4 shows that the performance of SNL drops off at a slower rate than PMMH with increasing $N$, and is noticeably faster for the $N = 1,000$ and $2,000$ experiments.

Table 2: Comparison of SNL and PMMH posterior statistics for $R_0$ and $\kappa$ in the SEIAR model.

| Parameter | $N$ | M | | S | |
|---|---|---|---|---|---|
| | | avg | max | avg | max |
| $R_0$ | 350 | -0.003 | -0.02 | 0.04 | 0.08 |
| | 500 | 0.005 | 0.09 | -0.02 | -0.09 |
| | 1000 | 0.05 | 0.08 | 0.13 | 0.20 |
| | 2000 | 0.02 | 0.04 | 0.20 | 0.31 |
| $\kappa$ | 350 | 0.20 | 0.32 | 0.01 | 0.04 |
| | 500 | -0.08 | -0.20 | -0.06 | -0.14 |
| | 1000 | -0.16 | -0.29 | -0.0004 | 0.03 |
| | 2000 | -0.25 | -0.35 | 0.02 | 0.07 |

Table 3: C2ST scores and p-values for the SEIAR model experiments.

| $N$ | Classifier Score | p-value |
|---|---|---|
| 350 | 0.4996 | 0.9514 |
| 500 | 0.5031 | 0.6670 |
| 1000 | 0.4975 | 0.7235 |
| 2000 | 0.5127 | 0.0834 |

### 4.3 Predator-Prey model

Here, we consider data generated from the predator-prey model described in McKane & Newman (2005), which exhibits resonant oscillations in the predator and prey populations. The model is a CTMC with state $(P_t, Q_t)$ where $P_t$ and $Q_t$ are the numbers of predators and prey at time $t$ respectively. The model has 5 parameters, $b, d_1, d_2, p_1, p_2 > 0$, which are the prey birth rate, predator death rate, prey death rate, and two predation rates respectively. The task is to infer these 5 parameters from predator and prey counts collected at discrete time steps, when any individual predator/prey has a probability $r$ (assumed to be fixed and known) of being counted. We use the values 0.5, 0.7 and 0.9 for $r$, and all noisy observations are generated from a single underlying realisation, where we simulate the predator and prey populations over 200 time units, recording the population every 2 time units; see Appendix C for a plot of the data with $r = 0.9$. We order the predators before the prey in the autoregressive model conditionals in Equation 6. For these experiments, we only use a simulation budget of $25,000$, because the observations are longer and more informative than in previous sets of experiments.

Table 4 shows the comparison between the posterior statistics for the experiments with $r = 0.9$, which suggests negligible differences between the posterior mean and standard deviations. It is also worth noting that SNL correctly reproduces a highly correlated posterior between $d_1$ and $p_1$. Furthermore, the $d_2$ posterior

is almost indistinguishable from the prior, suggesting that the autoregressive model can learn to reparameterise $\boldsymbol{\theta}$ into a set of identifiable parameters and suppress dependence on unidentifiable parameters. The accuracy of SNL increases with increasing $r$ for this model (see Appendix C for the other metrics), which is unsurprising, as there is a higher 'signal-to-noise ratio' in the training data for larger $r$, which leads to training data which is more informative for $p(\boldsymbol{y}_{1:n}|\boldsymbol{\theta})$. This is in direct contrast to PMMH, where more noise makes the particle filter less prone to weight degeneracy. The right of Figure 4 shows that SNL is an order of magnitude more efficient than PMMH for these experiments, and the ESS/s does not drop off with increasing $r$.

Table 4: Comparison of SNL and PMMH posterior statistics in the predator prey model with $r = 0.9$.

| Parameter | M | | S | |
|---|---|---|---|---|
| | avg | max | avg | max |
| $b$ | -0.03 | -0.07 | -0.04 | 0.11 |
| $d_1$ | 0.01 | 0.03 | -0.002 | -0.05 |
| $d_2$ | -0.002 | -0.12 | 0.01 | 0.10 |
| $p_1$ | 0.01 | 0.02 | -0.006 | -0.05 |
| $p_2$ | -0.05 | -0.10 | -0.05 | -0.08 |

## 5   Discussion

In this paper we have constructed a surrogate likelihood model for integer valued time series data. Such data often arises from counting processes and models of small populations and presents particular challenges for performing inference due to the low noise in the observation process. Low noise leads to highly informative observations of parts of the state, which means that a naive application of simulation based inference methods usually fails; more sophisticated approaches, typically utilising bespoke importance sampling simulations, do work, but are computationally expensive and difficult to implement. The main feature of our neural model is that it can be trained using simple unconditional simulations of the dynamical model. Unconditional simulation is both much easier to implement and faster to run and our experiments demonstrated significant reduction in the overall computational expense of performing inference in scenarios where exact sampling methods struggle.

Our method produces good approximations of the parameter posterior, accurately reproducing many features of the posteriors obtained from exact sampling methods. There are negligible biases in the posterior means across experiments, with the exception of the parameter $\kappa$ in the SEIAR model. This parameter has a very weak relationship with the data except near $\kappa = 0$ or 1, so it is unsurprising that it is difficult to learn. Our method always yielded posterior standard deviations which were either very close to that of PMMH, or slightly too large, suggesting that we generally avoid overconfident estimates, which can be a problem in simulation based inference (Hermans et al., 2022). Furthermore, an approximate posterior that has slightly inflated variance can still be used as an importance sampling distribution in a more computationally expensive exact method, which can significantly improve efficiency (Papamakiros & Murray, 2015) Our method is also relatively robust to randomness within the algorithm, not showing any major differences over independent runs of SNL.

In the right-skewed $R_0$ posteriors in the SEIAR model experiments, we saw that the SNL posteriors were slightly heavy in the tail. Improved training of the likelihood in these tail regions requires a scheme for generating SNL proposals that better cover these, e.g. by jittering the proposed samples. A particularly simple alteration would be to use the geometric mean of the SNL posterior and the prior, which leads to more sampling in the tails and better performance in sequential ABC methods (Alsing et al., 2018). Bayesian active learning has also been employed in SNL before (Lueckmann et al., 2018), and this kind of approach has the advantage that it allows for assessing the uncertainty in the surrogate likelihood, so it might be possible to derive an uncertainty score for the approximate posterior samples based on the surrogate uncertainty. In general, more robust SNL training routines are an important topic of current research (Kelly et al., 2024).

Neural density estimators have been the topic of research for some time now (Germain et al., 2015; Papamakarios et al., 2017). The work most similar to ours is described in the original SNL paper Papamakarios et al. (2019), but a limitation of that is the use of summary statistics (as in an ABC approach) for training the model. Instead, our autoregressive approach allows us to fit directly to the raw time series data, not just statistics derived from it. This is important for our applications involving very small populations (households) where it is difficult to form meaningful statistics due to the length and sparsity of the data (Walker et al., 2017). The convolutionial architecture of our model also means that the approximate likelihood can be evaluated in a single pass, rather than the sequential evaluation of a recurrent based architecture which is slower (van den Oord et al., 2016; Vaswani et al., 2017).

A neural surrogate for the likelihood has a number of additional advantages over other simulation based inference approaches such as PMMH. Our likelihood approximation is differentiable with respect to the parameters of the model, so a fast Hamiltonian Monte Carlo (HMC) kernel (Neal, 2011) can be used rather than a naive random walk MCMC kernel. Our method also has specific advantages over PMMH for problems involving longer time series or tall data, i.e. large numbers of iid observations Bardenet et al. (2017), such as seen in Section 5.1. The performance of PMMH depends strongly on the variance of the likelihood estimate (Doucet et al., 2015; Sherlock et al., 2015), with the chain becoming 'sticky' if this is too high. The variance grows linearly with the number of observations, but in practice this will lead to a quadratic growth in the computational expense of running the filter as more simulated particles are needed for the additional observations and the number of particles for all observations must also be increased to reduced the variance of each. Our neural approximation is not an estimate, so does not suffer this problem, but because it is only an approximation it may introduce a bias in the posterior samples.

In practice, we found that some care is needed in choosing the support of the priors. For example, in the predator prey experiments, without truncating the tails of the log normal priors sometimes the MCMC chain would end up stuck in non-physical regions of parameter space. For these non-physical parameter values, the simulated realisations are more or less trivial (extinction/saturation occurs very quickly), so presumably the surrogate has difficulty in learning the behaviour of the observations in these regions of the parameter space. Many SBI methods can suffer from similar problems if the simulated data and observation are not similar enough. Truncating the priors then works well to solve this issue because it prevents the surrogate from needing to be evaluated in these non-physical regions. Prior predictive checks are a standard way of checking the validity of priors and can also help diagnose these issues (Gelman, 2014). Additionally, physical interpretations of the parameters (as in the predator prey example) can often be used to find reasonable bounds for the model parameters.

A potential drawback of our method on time series with a large number of features is that the autoregressive model requires an ordering in the features. Such an ordering may not be natural and therefore could require significant amounts of tuning. Furthermore, in general the number of hidden channels in the CNN grows linearly with the input dimension, so the computational complexity of evaluation generally grows quadratically with the input dimension. Choosing a feature ordering is an issue for the model MADE as well (Germain et al., 2015), and the authors address this by randomly permuting the feature orderings and creating an ensemble, so a similar technique could be employed for our model. We also point out that for many physical models choosing a feature ordering is unlikely to be a problem. In epidemiological settings, multidimensional time series of counts would be rare, and for other dynamical systems of interest there is often additional causal structure that creates an obvious choice of ordering. Furthermore, for high dimensional data, it is likely the case that a continuum approximation can be employed, or the use of summary statistics becomes a viable choice, and in both cases the method developed in this paper would not be required.

This work assumes that conditioning a finite number of steps back in time is sufficient to model the conditionals. If longer range dependencies were important, then dilated causal convolutions (van den Oord et al., 2016) or self attention (Vaswani et al., 2017) could be employed to construct the autoregressive model. Self attention could also be beneficial for high dimensional time series, since it has linear complexity in the number of hidden features compared to the quadratic complexity of convolution layers. Another line of enquiry would be to develop better diagnostics to assess the results of SNL. The diagnostics used in Papamakarios et al. (2019) did not fit well with our integer valued data, nor did it fit well with the random length time series in the epidemic models. Studying how robust our method is to model misspecification would also

be important future work, as any real data is unlikely to truly follow a specific model (Kelly et al., 2024). Finally, one could use a family of distributions other than the DMoL to represent the surrogate likelihood conditionals. For example, the household model experiments in Section 4.1 could possibly be treated by a simple multinomial/categorical distribution. This works because the population sizes are small enough so that the observations $y_i$ can only take a few possible values, which can be treated as distinct categories.

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

# A   Details of Autoregressive Model

## A.1   One dimensional inputs

A causal convolution layer $\ell : \mathbb{R}^{n \times d_{\text{in}}} \to \mathbb{R}^{n \times d_{\text{out}}}$ with convolution kernel $(W_1, \cdots, W_k)$ (where $W_j \in \mathbb{R}^{d_{\text{out}} \times d_{\text{in}}}$) and bias $\text{b} \in \mathbb{R}^{d_{\text{out}}}$ is given by

$$\ell(\mathbf{z}_{1:n})_i = \sum_{j=0}^{k-1} \mathbf{1}_{i>j} W_{k-j}\, \mathbf{z}_{i-j} + \mathbf{b},$$

To make the layer $\boldsymbol{\theta}$-dependent, we augment it with an additive transformation of $\boldsymbol{\theta}$. We first calculate a context vector from $\boldsymbol{\theta}$ using a shallow neural network (with activation $\sigma$)

$$\mathbf{c}(\boldsymbol{\theta}) = W_{\mathbf{c}}^2 \sigma(W_{\mathbf{c}}^1 \boldsymbol{\theta} + \mathbf{b_c}). \tag{7}$$

We then use a linear transformation of $\mathbf{c}(\boldsymbol{\theta})$ to add $\boldsymbol{\theta}$ dependence to $\ell$:

$$\ell(\mathbf{z}_{1:n}, \mathbf{c}(\boldsymbol{\theta}))_i = \sum_{j=0}^{k-1} \mathbf{1}_{i>j} W_{k-j}\, \mathbf{z}_{i-j} + + W_{\boldsymbol{\theta}} \mathbf{c}(\boldsymbol{\theta}) + \mathbf{b}\,. \tag{8}$$

The size of the hidden layer and the output in Equation 7 are both hyperparameters.

Using an activation function $\sigma : \mathbb{R} \to \mathbb{R}$, a residual block in the CNN is given by

$$\ell_{\text{res}}(\mathbf{z}_{1:n}, \mathbf{c}(\boldsymbol{\theta}))_i = \mathbf{z}_i + \ell^{(2)} \left( \sigma \left( \ell^{(1)} \left( \sigma(\mathbf{z}_{1:n}), \mathbf{c}(\boldsymbol{\theta}) \right) \right), \mathbf{c}(\boldsymbol{\theta}) \right)_i, \tag{9}$$

where $\ell^{(1)}$ and $\ell^{(2)}$ are of the form in Equation 8, both having the same number of input/output channels for simplicity.

From Equation 8, it can be seen that $\ell(\mathbf{z}_{1:n}, \mathbf{c}(\boldsymbol{\theta}))_i$ is dependent on $\mathbf{z}_{i-(k-1):i}$. Defining $f(y_{1:n}, \boldsymbol{\theta})$ as the composition of $h$ causal convolutions[4], it follows that $f(y_{1:n}, \mathbf{c}(\boldsymbol{\theta}))_i$ is dependent on $y_{i-h(k-1):i}$. Evaluation of the CNN $\mathcal{N}_\psi(y_{1:n}, \boldsymbol{\theta})$ and calculation of the logistic parameters from the output is given in Algorithm 1. Note that we assume that the initial observation $y_0$ is known, so that the shifts of $q_\psi(y_1 | \boldsymbol{\theta}, y_0)$ can be calculated in the same form as the other shifts.

## A.2   Multidimensional inputs

Assume $\boldsymbol{y}_i$ has $d$ components, and for simplicity assume that the output of each residual block has $dp$ output channels, where $p$ is a positive integer. For $j = 1, 2$, define the following $d \times d$ block matrix

$$M_j = \begin{bmatrix} 1_{p \times d_j} & 0_{p \times d_j} & \cdots & 0_{p \times d_j} \\ 1_{p \times d_j} & 1_{p \times d_j} & \cdots & 0_{p \times d_j} \\ \vdots & \ddots & & \vdots \\ 1_{p \times d_j} & 1_{p \times d_j} & \cdots & 1_{p \times d_j} \end{bmatrix},$$

where $0_{m \times n}/1_{m \times n}$ are $m \times n$ matrices of ones/zeros and $d_1 = 1$, $d_2 = p$. In other words, $M_j$ is a block lower triangular matrix where all entries on and below the block diagonal are 1. The masked causal convolution appropriate for the hidden layers of the CNN is then defined by

$$\tilde{\ell}(\mathbf{z}_{1:n}, \boldsymbol{\theta}) = W_{\boldsymbol{\theta}} \mathbf{c}(\boldsymbol{\theta}) + (M_j \odot W_k) \boldsymbol{y}_k + \sum_{l=1}^{k-1} W_{k-l}\, \mathbf{z}_{i-l} + \text{b}, , \tag{10}$$

where $j = 1$ for the first layer, and $j = 2$ for the causal convolutions within residual blocks. The mask ensures that the first block of $p$ channels in the hidden layers of the CNN is only dependent on $y_i^1$, the

---

[4]The composition can include residual blocks, where one residual block counts for two causal convolutions

---

**Algorithm 1** Evaluation of $\mathcal{N}_\psi(y_{1:n}, \boldsymbol{\theta})$

---

**Input**: Time series with initial value $y_{0:n}$, first CNN layer $\ell$, $r$ residual blocks $\ell_{\text{res}}^{(1)}, \ldots, \ell_{\text{res}}^{(r)}$, final CNN layer $\ell_{\text{f}}$, neural network parameters $\psi$.
**Output**: Sequence of shift, scale and mixture proportions for a mixture of $c$ discretised logistic conditionals.
1: Calculate context $\mathbf{c} \leftarrow W_{\mathbf{c}}^2 \sigma(W_{\mathbf{c}}^1 \boldsymbol{\theta} + \mathrm{b}_{\mathbf{c}})$.
2: Apply first layer $\mathbf{z}_{1:n} \leftarrow \ell(y_{1:n}, \mathbf{c})$.
3: **For** $k = 1, \ldots, r$ **do**
4:     Apply the $k^{\text{th}}$ residual block $\mathbf{z}_{1:n} \leftarrow \ell_{\text{res}}^{(k)}(\mathbf{z}_{1:n}, \mathbf{c})$.
5: Apply the final layer $\boldsymbol{o}_{1:n} \leftarrow \ell_{\text{f}}(\mathbf{z}_{1:n}, \mathbf{c})$.
6: Calculate the logistic parameters.
7: **For** $i = 2, \ldots, n$ **do**
8:     **For** $j = 1, \ldots, c$ **do**
9:         Calculate the shifts $\mu_i^j \leftarrow \boldsymbol{o}_{i-1}^j + y_{i-1}$.
10:         Calculate the scales $s_i^j \leftarrow \mathrm{softplus}(\boldsymbol{o}_{i-1}^{j+c}) + 10^{-6}$.
11:         Calculate the mixture proportions $w_{i,j} \propto \exp(\boldsymbol{o}_{i-1}^{j+2c})$.
12: **Return** $\boldsymbol{\mu}_{2:n}$, $\boldsymbol{s}_{2:n}$, $\boldsymbol{w}_{2:n}$.

---

second block is dependent on $\left(y_i^1, y_i^2\right)$, etc. In other words, if $\mathrm{z}_{1:n}$ is the output of any hidden layer, then the elements $\mathrm{z}_i^{1+(j-1)p}, \ldots, \mathrm{z}_i^{jp}$ are only dependent on $\boldsymbol{y}_i$ through the elements $y_i^1, \ldots, y_i^j$. The CNN will output $3cd$ output channels, where each block of $d$ channels corresponds to $3c$ mixture of logistic parameters. To remove dependence of the $j^{\text{th}}$ block of output channels on $y_i^j$, we apply a $d \times d$ block lower triangular mask matrix similar to $M_j$ to the final layer, but where the block diagonal set to zeros:

$$
M = \begin{bmatrix} 0_{3c \times p} & 0_{3c \times p} & \cdots & 0_{3c \times p} \\ 1_{3c \times p} & 0_{3c \times p} & \cdots & 0_{3c \times p} \\ \vdots & \ddots & & \vdots \\ 1_{3c \times p} & 1_{3c \times p} & \cdots & 0_{3c \times p} \end{bmatrix}.
\tag{11}
$$

The same procedure holds when $\boldsymbol{y}_i$ contains categorical components, though number of output channels and the block sizes in the mask Equation 11 will need to be adjusted so that the correct number of parameters are obtained to model the categorical distributions rather than DMoLs.

## B   Additional Experiment Details

### B.1   Hyperparameters and Implementation Details

All experiments were performed using a single NVIDIA RTX 3060TI GPU, which provides a significant speed up over a CPU due to the use of convolutions. All instances of PMMH were written in the Julia programming language, and the particle filters were run on 16 threads to improve performance.

We used a kernel length of 5 for all experiments, and since all observations have relatively low to no noise, this was sufficient to ensure that $p(\boldsymbol{y}_i | \boldsymbol{y}_{i-m:i-1}, \boldsymbol{\theta}) \approx p(\boldsymbol{y}_i | \boldsymbol{y}_{1:i-1}, \boldsymbol{\theta})$ for all experiments. We used 5 mixture components in all experiments. We found that more mixture components resulted in slower training without a significant increase in performance, and less would slightly worsen accuracy without much increase in speed. For the SIR/SEIAR model experiments, we used 64 hidden channels and 2/3 residual blocks. For the predator-prey model experiments, we used 100 hidden channels and 3 residual blocks. All of these hyperparameters were chosen initially to be relatively small, and were increased when necessary based on an pilot run of training, adding depth to the network over increasing hidden channels, since computational cost grows quadratically with the number of hidden channels. We also used the same activation function for all experiments, namely the Gaussian linear error unit (GeLU) (Hendrycks & Gimpel, 2016).

For all experiments, we used 90% of the data for training and the other 10% for validation. We used Adam with weight decay (Loshchilov & Hutter, 2017) as the optimiser, using a weight decay parameter of $10^{-5}$ and a learning rate of 0.0003. The only other form of regularisation which we used was early stopping, using a patience parameter of 30 for all experiments. For the SIR/SEIAR/predator-prey model experiments, we used a batch size of 1024/256/512 to calculate the stochastic gradients. We used a large batch size for the SIR model experiments, because in general each observation was quite short and therefore sparse of data. For the SEIAR model experiments, the autoregressive model was prone to getting stuck at a local minimum in the first round of training with too high a batch size, so presumably the stochasticity in the gradients helped to break through the local minimum.

For the SIR model experiment with a single observation and the SEIAR model experiments, we used a simulation budget of 60,000 samples, using 10,000 from the prior predictive in the first round, hence we drew 5,000 MCMC samples per round. For the household experiments, we used a simulation budget of 100,000 due to the shorter observations, hence we drew 9,000 MCMC samples per round. For the predator-prey model, we used a simulation budget of 25,000, using 10,000 from the prior predictive, hence we drew 1,500 MCMC samples per round. We dropped the simulation budget in the predator prey experiments, because the time series were much longer on average, and MCMC more expensive for this data. Additionally, we always drew an extra 200 samples at the beginning of MCMC, which were used in the warmup phase to tune the hyperparameters of NUTS, and then were subsequently discarded.

For the classifier 2-sample t-test values, we use the same procedure employed by Lueckmann et al. (2021), namely we use a two layer network with 50 hidden units (10 times the 5 SEIAR model parameters), a ReLU activation, and a single output unit, which after applying a sigmoid function gives the probability that the sample comes from the true (PMMH) posterior. To generate the training data, we used one instance of SNL and generated an ESS of approximately 10,000 using PMMH, thinning so that the number of raw PMMH and SNL samples are approximately equal. We randomly permuted the full dataset, and saved 5000 samples to calculate the classifier score, while 10% of the remaining samples were used for validation and the rest for training. After training the classifier using the Adam optimiser, we evaluate the classifier on the 5000 samples set aside and recorded their average. We performed this 10 times with a different random seed to account for biases that might have been introduced through the dataset splits, reporting the average over these 10 instances for each experiment. The p-values reported were calculated by taking the classifier score $s$ and calculating $z = 2\sqrt{5000}(s - 0.5)$, which is a standard normal random variable under the hypothesis that the two distributions are the same (Lopez-Paz & Oquabe, 2017), so the p-value is simply $\Pr(|Z| > z)$ where $Z \sim \mathrm{N}(0,1)$.

## B.2 SIR Model

### B.2.1 The Model

The SIR model is a CTMC which models an outbreak in a homogeneous population of fixed size $N$, classing each individual as either susceptible, infectious or removed (Allen, 2017). The model has state space

$$\left\{(S, I, R) \in \mathbb{Z}^3 \mid S, I, R \geq 0, \ S + I + R + N\right\},$$

and transition rates

$$(S, I, R) \rightarrow (S - 1, I + 1, R) \quad \text{at rate } \frac{\beta S I}{N - 1}, \tag{12}$$

$$(S, I, R) \rightarrow (S, I - 1, R + 1) \quad \text{at rate } \gamma I, \tag{13}$$

where $\beta > 0$ is the transmissibility and $\gamma > 0$ is the recovery rate. The SIR model can be recast in terms of two counting processes by using the transformations

$$Z_1 = N - S, \tag{14}$$

$$Z_2 = R, \tag{15}$$

where $Z_1$ and $Z_2$ count the number of times Equation 12 and Equation 13 occur respectively. The model parameters $(\beta, \gamma)$ can be reparameterised into more interpretable quantities $R_0 = \beta/\gamma$ and $1/\gamma$, where

the former is the average number of infections caused by a single infectious individual (when most of the population is susceptible), and the latter is the mean length of the infectious period (Allen, 2017).

### B.2.2 Autoregressive Model Tweaks

We assume that there is one initial infection, and that we subsequently observe the number of new cases every day (without any noise), equivalent to observing $Z_1$ at a frequency of once per day. We also assume that the outbreak has reached completion, hence the final size of the outbreak $N_F$ can be measured. Using the data $(y_{1:n}, n_F)$, we can calculate the day at which the final size is reached:

$$\tau = \min\{i | y_i = n_F\}.$$

Clearly,

$$p(y_{1:n}, N_F = n_F | \boldsymbol{\theta}) = p(y_{1:\tau}, N_F = n_F | \boldsymbol{\theta}),$$

since case numbers can never grow beyond the final size. We can take this further and define indicator variables

$$f_i = \delta_{y_i, n_F},$$

from which it is easy to see that

$$p(y_{1:\tau}, N_F = n_F | \boldsymbol{\theta}) = p(y_{1:\tau}, f_{0:\tau} | \boldsymbol{\theta}),$$

where we need to include $f_0$ in case the outbreak never progresses from the initial infection. In the context of the autoregressive model, we decompose the conditionals $p(y_i, f_i | y_{1:i}, f_{1:i}, \boldsymbol{\theta})$ as follows

$$\begin{aligned} p(y_i, f_i | y_{1:i}, f_{0:i}, \boldsymbol{\theta}) &= p(f_i | y_{1:i}, f_{0:i-1}, \boldsymbol{\theta}) p(y_i | y_{1:i-1}, f_{0:i-1}, \boldsymbol{\theta}), \\ &= p(f_i | y_{1:i}, f_{i-1}, \boldsymbol{\theta}) p(y_i | y_{1:i-1}, f_{i-1}, \boldsymbol{\theta}), \end{aligned}$$

since $f_{i-1} = 0$ implies that $f_k = 0$ for all $k < i-1$. Furthermore, since $f_i = 1$ if and only if $i = \tau$, then it follows that $f_{i-1} = 0$ for all $i \le \tau$. It would therefore be redundant to make $q_\psi(y_i, f_i | y_{1:i-1}, f_{i-1}, \boldsymbol{\theta})$ explicitly dependent on $f_{i-1}$, since $f_{i-1} = 0$ always by construction. As a result, we only need to use $y_{1:\tau}$ as an input to the CNN. We do require a tweak to the output layer of the CNN, so that it outputs 16 parameters per conditional, 15 for the DMoL parameters, and one for $q_\psi(f_i | y_{1:i}, f_{i-1}, \boldsymbol{\theta})$. Since $f_i$ should be conditioned on $y_i$, we use a mask on the final causal convolution layer which has a similar structure to Equation 11 in Appendix A. Explicitly, the mask matrix has 15 rows of zeros, followed by a single row of ones, ensuring that only the parameter corresponding to $q_\psi(f_i | y_{1:i}, f_{i-1}, \boldsymbol{\theta})$ is dependent on $y_i$, and the other 15 are not.

In practice, we do not need to include $f_0$ in our calculation with the autoregressive model, as $p(f_0 | \boldsymbol{\theta})$ can be calculated analytically using the rates Equation 12, Equation 13 (Karlin & Taylor, 1975):

$$p(f_0 | \boldsymbol{\theta}) = \frac{f_0 + (1 - f_0) R_0}{R_0 + 1}. \tag{16}$$

We therefore train the autoregressive model on samples conditioned on $N_F > 1$, and define the surrogate likelihood to be

$$q_\psi(y_{1:n}, n_F | \boldsymbol{\theta}) \equiv p(f_0 | \boldsymbol{\theta}) q_\psi(y_{1:\tau}, f_{1:\tau} | \boldsymbol{\theta}, f_0 = 0).$$

This prevents us from wasting simulations on time series with 0 length, which are likely when $R_0$ is small. In principle, we could also calculate $p(N_F > t | \boldsymbol{\theta})$ for some threshold size $t$, and only train on simulations which cross this threshold, which would further reduce the number of short time series in the training data. This is possible largely due to the simplicity of the SIR model, and such a procedure generally becomes more difficult for CTMCs with a higher dimensional state space.

The final augmentation that we make to the autoregressive model concerns masking out certain terms of the form $\log q_\psi(f_i | y_{1:i}, f_{i-1}, \boldsymbol{\theta})$, which forces $q_\psi(f_i | y_{1:i}, f_{i-1}, \boldsymbol{\theta}) = p(f_i | y_{1:i}, f_{i-1}, \boldsymbol{\theta}) = 1$. There are three scenarios where this is appropriate:

1. If $y_i = y_{i-1}$ and $f_i = 0$, as we know that $N_F \ne y_{i-1}$, and the outbreak does not process over the interval $[i-1, i]$, so $f_i = 0$ with probability 1.

2. If $y_1 = 1$, since we always have $N_F > 1$ by construction.

3. If $y_i = N$, since this corresponds to the whole population becoming infected, therefore we have $N_F = N$ and $f_i = 1$ with probability 1.

We calculate the context vector $\mathbf{c}((\beta, \gamma))$ using a shallow neural network with 64 hidden units and 12 output units, using a GeLU non-linearity. For the household experiments, we additionally concatenate a one-hot encoding of the household size to $(\beta, \gamma)$ before calculating $\mathbf{c}$, so that the autoregressive model can distinguish between household sizes. To obtain the logistic parameters from the output of the neural network $\boldsymbol{o}_{1:\tau}$, we calculate[5]

$$\mu_i^j = o_{i-1}^j N + (1 - o_{i-1}^j) y_{i-1},$$
$$s_i^j = (N - y_{i-1})\text{softplus}(o_i^{j+5}) + \epsilon.$$

This encourages the shifts to lie in the support of the conditional $\{y_{i-1}, \dots, N\}$ (since $y_i$ is increasing), and allows the scales to be learnt in proportion to the width of the support of each conditional.

### B.2.3 Training and Inference Details

For inference, we used a Uniform$(0.1, 10)$ prior on $R_0$ and a Gamma$(10, 2)$ prior on $1/\gamma$, where we truncate the prior of $1/\gamma$ so that it is supported on $[1, \infty)$. For the single observation experiment, we simulate the data using the Gillespie (1976) algorithm, running for a maximum length of 100 days, which ensures that approximately 99.9% of samples (estimated based on simulation) from the prior predictive reach their final size. We construct a binary mask for the simulations where $\tau < 100$ so that values following $\tau$ do not contribute to the loss during training.

For the household experiments, we did not need to generate training data from households of size 2, as the likelihood of such observations can be calculated analytically. The case of $N_F = 1$ is handled by Equation 16, and if $N_F = 2$ it can be shown that

$$p(y_{1:\tau}, n_F = 2 | \boldsymbol{\theta}, N_F > 1) = e^{-\tau(\beta+\gamma)} \left(1 - e^{-(\beta+\gamma)}\right).$$

The training data was generated from five realisations for a single value of $\boldsymbol{\theta}$, using a unique household size (between 3 and 7) for each of the realisations. We found that it was useful to increase the variance of the proposals generated from MCMC, as the proposals had small variance and sometimes this seemed to reinforce biases over the SNL rounds. We inflated the variance of the proposals by fitting a diagonal covariance Gaussian to the proposed samples (centred to have mean 0), and added noise from the Gaussian to the proposals, effectively doubling the variance of the proposals over the individual parameters.

For all experiments on the SIR model we, we used the particle filter given in Black (2019) within PMMH. For the single observation experiment, we used 20 particles, and the MCMC chain was run for 30,000 steps, which was sufficient for this experiment since the particle filter works particularly well on the SIR model. For the household experiments, we used 20, 40 and 60 particles for 100, 200 and 500 households respectively. We ran the MCMC chain for long enough as necessary to obtain an ESS of approximately 1000 or above for both parameters. Mixing of the MCMC chain slows down with increasing household numbers due to increasing variance in the log-likelihood estimate that must be offset by also increasing the number of particles. The covariance matrix for the Metropolis-Hastings proposal was tuned based on a pilot run for all experiments.

### B.3 SEIAR Model

This model is another outbreak model, and is described in Black (2019). It is a CTMC with state space

$$\{(Z_1, Z_2, Z_3, Z_4, Z_5) \mid Z_1 \geq Z_2 \geq Z_3 \geq Z_4 \geq 0, \ Z_1 \geq Z_5 \geq 0, \ Z_1 + \dots + Z_5 = N\},$$

---

[5]Note that we input $y_{0:\tau}$ into the neural network so we can calculate $q_\psi(y_1 | \boldsymbol{\theta})$.

and transition rates

$$(Z_1, Z_2, Z_3, Z_4, Z_5) \rightarrow (Z_1 + 1, Z_2, Z_3, Z_4, Z_5) \quad \text{at rate } \frac{(N - Z_1)(\beta_p(Z_2 - Z_3) + \beta_s(Z_3 - Z_4))}{N}, \quad (17)$$

$$(Z_1, Z_2, Z_3, Z_4, Z_5) \rightarrow (Z_1, Z_2 + 1, Z_3, Z_4, Z_5) \quad \text{at rate } q\sigma(Z_1 - Z_2 - Z_5), \quad (18)$$

$$(Z_1, Z_2, Z_3, Z_4, Z_5) \rightarrow (Z_1, Z_2, Z_3 + 1, Z_4, Z_5) \quad \text{at rate } \gamma(Z_2 - Z_3), \quad (19)$$

$$(Z_1, Z_2, Z_3, Z_4, Z_5) \rightarrow (Z_1, Z_2, Z_3, Z_4 + 1, Z_5) \quad \text{at rate } \gamma(Z_3 - Z_4), \quad (20)$$

$$(Z_1, Z_2, Z_3, Z_4, Z_5) \rightarrow (Z_1, Z_2, Z_3, Z_4, Z_5 + 1) \quad \text{at rate } (1 - q)\sigma(Z_1 - Z_2 - Z_5). \quad (21)$$

The event Equation 17 represents a new exposure, Equation 18 represents an exposure becoming infectious (but pre-symptomatic), Equation 19 represents a pre-symptomtic individual becoming symptomatic, Equation 20 represents a recovery, and Equation 21 represents an exposure who becomes asymptomatic (and not infectious). The parameters of the model are the transmissibilities attributed to the pre-symptomatic and symptomatic population $\beta_p, \beta_s > 0$, the rate of leaving the exposed class $\sigma > 0$, the rate of leaving the (pre)symptomatic infection class $\gamma > 0$, and the probability of becoming infectious $q \in (0, 1)$. We follow Black (2019) and reparameterise the model in terms of the parameters $R_0 = (\beta_p + \beta_s)/q\gamma$, $1/\sigma$, $1/\gamma$, $\kappa = \beta_p/(\beta_p + \beta_s)$ and $q$. We use the same parameters to generate the observations and priors that were used in Black (2019): $R_0 = 2.2$, $1/\gamma = 1$, $1/\sigma = 1$, $\kappa = 0.7$ and $q = 0.9$, and priors

$$R_0 \sim \text{Uniform}(0.1, 8),$$
$$\frac{1}{\sigma} \sim \text{Gamma}(10, 10),$$
$$\frac{1}{\gamma} \sim \text{Gamma}(10, 10),$$
$$\kappa \sim \text{Uniform}(0, 1),$$
$$q \sim \text{Uniform}(0.5, 1),$$

truncating the priors for $1/\sigma$ and $1/\gamma$ to have support $[0.1, \infty)$ and $[0.5, \infty)$ respectively.

The observations are given by the values of $Z_3$ at discrete time steps, corresponding to the number of new symptomatic individuals per day, as well as the final size $N_F$. We handle the final size data in the same way as discussed for the SIR model, but we include $f_0$ for this model, as $p(N_F = 1|\boldsymbol{\theta})$ is more difficult to calculate due to the asymptomatic individuals. The initial state is taken to be $(Z_1, Z_2, Z_3, Z_4, Z_5) = (1, 1, 0, 0, 0)$, which we simulate forward using the Gillespie algorithm until $Z_3 = 1$, after which we begin recording the values of $y_i$ at integer time steps. For population sizes of 350, 500, 1000 and 2000, we generate time series of maximum length 70, 80, 100 and 110 respectively. This ensured that at least 99% of samples from the prior predictive reached their final size (estimated by simulation). The autoregressive model follows the exact same construction as given for the SIR model experiments, except that we rescale the time series by the population size before inputting to the CNN. We also divide $y_{1:\tau}$ by $N$ before inputting to the CNN, which helps to stabilise training.

For PMMH, we used the particle filter given in Black (2019), using 70, 80, 100 and 120 particles for population sizes of 350, 500, 1000 and 2000 respectively. The MCMC chain was run until all parameters had an ESS of approximately 1000 or above, and the proposal covariance matrix was estimated based on a pilot run.

### B.4   Predator-Prey Model

The predator-prey model that we use is described in McKane & Newman (2005). For large populations in predator-prey models, it is often sufficient to consider a deterministic approximation via an ODE. For this model, the corresponding ODE has a stable fixed point, which makes it incapable of exhibiting the cyclic behaviour seen in e.g. the Lotka-Volterra model (Wilkinson, 2018). In the stochastic model however, there is a resonance effect which causes oscillatory behaviour. Unlike predator-prey models with limit cycle behaviour, in this model the dominant frequencies in the oscillations have a random contribution to the trajectory, which causes greater variability among the realisations for a fixed set of parameters.

The model is a CTMC which represents the population dynamics of a predator population of size $P$ and a prey population of size $Q$. The state space given by

$$\left\{ (P,Q) \in \mathbb{Z}^2 \mid 0 \le P, Q \le K, \; P + Q \le K \right\},$$

where $K \ge 0$ is a carrying capacity parameter. The transition rates are given by

$$(P,Q) \to (P-1,Q) \quad \text{at rate } d_1 P, \tag{22}$$

$$(P,Q) \to (P,Q+1) \quad \text{at rate } 2b\frac{Q}{K}(K-P-Q), \tag{23}$$

$$(P,Q) \to (P,Q-1) \quad \text{at rate } 2p_2\frac{PQ}{K} + d_2 Q, \tag{24}$$

$$(P,Q) \to (P+1,Q-1) \quad \text{at rate } 2p_1\frac{PQ}{K}, \tag{25}$$

where $b, d_1, d_2, p_1, p_2 > 0$ are the parameters of the model. To simulate the data, we use parameters $b = 0.26$, $d_1 = 0.1$, $d_2 = 0.01$, $p_1 = 0.13$, $p_2 = 0.05$. We use informative priors on $b$, $d_1$ and $d_2$, assuming that reasonable estimates can be obtained based on observing the two individual species. In particular we use

$$\log b \sim N(\log 0.25, 0.25),$$
$$\log d_1 \sim N(\log 0.1, 0.5),$$
$$\log d_2 \sim N(\log 0.01, 0.5),$$

truncating each prior to have support $(-\infty, 0]$ in the log domain. The effect of this truncation is minimal: the widest prior, for $\log(b)$, has upper bound $-1.22$ for the 99% confidence interval. For the two predation parameters, we use priors

$$p_1 \sim \text{Exponential}(0.1),$$
$$p_2 \sim \text{Exponential}(0.05),$$

where we truncate the prior of $p_1$ so that it is supported on $[0.01, \infty)$. This truncation, which removes only 5.4% of the mass from $p_1$, rules out a population explosion in the prey due to unphysically low predation. Furthermore, when $p_1$ is close to 0, (25) becomes negligible and the resulting model does not behave like a true predator-prey model, so truncating the prior rules out this possibility. To simulate the realisation for the observations, we use a carrying capacity of $K = 800$ and initial condition $(250, 250)$. We then run the Gillespie algorithm over 200 time steps, taking $\boldsymbol{y}_i' = (P_{2i}, Q_{2i})$, and apply observation error to generate the observations, namely

$$y_i^1 \sim \text{Binomial}(y_{i,1}', r),$$
$$y_i^2 \sim \text{Binomial}(y_{i,2}', r),$$

for $r = 0.5, \; 0.7, \; 0.9$.

To generate the training data, we firstly infer the initial condition from the noisy initial observation $\boldsymbol{y}_0$. In particular, we assume $p(P_0) \propto 1$ and $p(Q_0) \propto 1$, from which it is easy to derive the posterior $p((P_0, Q_0)|\boldsymbol{y}_0)$:

$$\left( P_0 - y_0^1 \right) |\boldsymbol{y}_0 \sim \text{NegativeBinomial}(y_{0,1} + 1, r),$$
$$\left( Q_0 - y_0^2 \right) |\boldsymbol{y}_0 \sim \text{NegativeBinomial}(y_{0,2} + 1, r).$$

We then generate the training data in the same way we produced the observations. No tweaks are made to the autoregressive model, though we do divide $\boldsymbol{y}_{1:100}$ by 60 before inputting to the CNN, and multiply the shifts (before adding $\boldsymbol{y}_{i-1}$) and scales by 60. We do this because the standard deviation across the time axis for samples from the prior predictive was estimated to be approximately 60. We also tried rescaling the input/output by larger values, but this generally resulted in less stable training and slower MCMC iterations. We do not centre the inputs, because if $y_i^j = 0$ for multiple time steps in a row, then this implies that an extinction has likely occurred, hence zeroing the weights associated with $y_i^j$ in the first layer

should correspond to the model learning the dynamics of the individual species. If we centred the data, then extinctions would not correspond to zeroing the weights. We also attempted to order the prey before the predators in the autoregressive model, but found that there was not a large difference between the two orderings.

For PMMH, we used a bootstrap particle filter (Kitagawa, 1996), using 100, 200 and 250 particles for $r = 0.5$, $0.7$ and $0.9$ respectively. We used 180,000 MCMC iterations for each experiment, which produced an ESS of greater than 500 for all parameters. Generating a large ESS for this data is difficult due to the extremely high posterior correlation between $d_1$ and $p_1$, and the computational time per MCMC iteration is quite high for these experiments. We estimated the covariance matrix for the MCMC kernel from a prior SNL run, and tuned further based on a pilot run. Without the SNL run to inform the initial covariance matrix, tuning the MCMC proposal would likely take huge amounts of computational time, since a proposal with diagonal covariance would mix extremely slowly with the highly correlated parameters in the posterior.

## C  Additional Results

### C.1  Posterior Metrics

Table 5: Comparison of SNL and PMMH posterior statistics for the SIR model.

| Parameter | M | | S | |
|:---:|:---:|:---:|:---:|:---:|
| | avg | max | avg | max |
| $R_0$ | 0.00007 | -0.0034 | 0.036 | 0.068 |
| $1/\gamma$ | -0.029 | -0.087 | 0.019 | 0.073 |

Table 6: Comparison of SNL and PMMH posterior statistics for $1/\sigma$, $1/\gamma$ and $q$ in the SEIAR model.

| Parameter | $N$ | M | | S | |
|:---:|:---:|:---:|:---:|:---:|:---:|
| | | avg | max | avg | max |
| | 350 | 0.04 | 0.10 | 0.03 | 0.07 |
| $1/\sigma$ | 500 | 0.01 | 0.08 | -0.01 | -0.05 |
| | 1000 | 0.02 | 0.08 | 0.02 | 0.05 |
| | 2000 | -0.007 | -0.05 | 0.04 | 0.07 |
| | 350 | 0.07 | 0.20 | 0.03 | 0.06 |
| $1/\gamma$ | 500 | 0.08 | 0.30 | 0.03 | 0.08 |
| | 1000 | 0.05 | 0.17 | 0.03 | 0.06 |
| | 2000 | 0.04 | 0.10 | 0.03 | 0.10 |
| | 350 | -0.03 | -0.06 | 0.02 | 0.06 |
| $q$ | 500 | -0.05 | -0.12 | -0.02 | -0.12 |
| | 1000 | -0.07 | -0.12 | -0.02 | -0.06 |
| | 2000 | -0.02 | -0.07 | 0.06 | 0.08 |

Table 7: Comparison of SNL and PMMH posterior statistics predator prey model experiments.

| Parameter | $r$ | M | | S | |
|---|---|---|---|---|---|
| | | avg | max | avg | max |
| $b$ | 0.5 | -0.09 | -0.22 | -0.02 | -0.10 |
| | 0.7 | -0.05 | -0.11 | -0.009 | -0.06 |
| $d_1$ | 0.5 | -0.08 | -0.18 | -0.01 | -0.08 |
| | 0.7 | -0.003 | -0.02 | 0.03 | 0.08 |
| $d_2$ | 0.5 | -0.00001 | -0.14 | -0.08 | -0.16 |
| | 0.7 | 0.04 | 0.12 | 0.05 | 0.17 |
| $p_1$ | 0.5 | -0.06 | -0.13 | -0.01 | -0.09 |
| | 0.7 | -0.0003 | -0.01 | 0.03 | 0.08 |
| $p_2$ | 0.5 | 0.02 | 0.13 | 0.02 | 0.07 |
| | 0.7 | -0.06 | -0.08 | -0.11 | -0.17 |

## C.2 Runtimes

Table 8: ESS/s values from PMMH versus SNL for household data experiments. Recall that we report the ESS/s value for SNL by averaging the ESS over the final 5 rounds.

| $h$ | PMMH ESS/s | SNL ESS/s | | |
|---|---|---|---|---|
| | | avg | min | max |
| 100 | 2.33 | 3.67 | 3.37 | 4.00 |
| 200 | 0.70 | 3.28 | 3.03 | 3.44 |
| 500 | 0.25 | 2.38 | 2.25 | 2.57 |

Table 9: ESS/s values from PMMH versus SNL for SEIAR experiments. Recall that we report the ESS/s value for SNL by averaging the ESS over the final 5 rounds.

| $N$ | PMMH ESS/s | SNL ESS/s | | |
|---|---|---|---|---|
| | | avg | min | max |
| 350 | 1.94 | 2.55 | 2.24 | 2.96 |
| 500 | 1.65 | 1.78 | 1.60 | 1.90 |
| 1000 | 0.50 | 1.50 | 1.29 | 1.62 |
| 2000 | 0.20 | 1.23 | 1.08 | 1.39 |

Table 10: ESS/s values from PMMH versus SNL for predator-prey experiments. Recall that we report the ESS/s value for SNL by averaging the ESS over the final 5 rounds.

| $r$ | PMMH ESS/s | SNL ESS/s | | |
|---|---|---|---|---|
| | | avg | min | max |
| 0.5 | 0.034 | 0.33 | 0.29 | 0.37 |
| 0.7 | 0.027 | 0.51 | 0.48 | 0.56 |
| 0.9 | 0.015 | 0.45 | 0.34 | 0.50 |

## D    Additional Plots

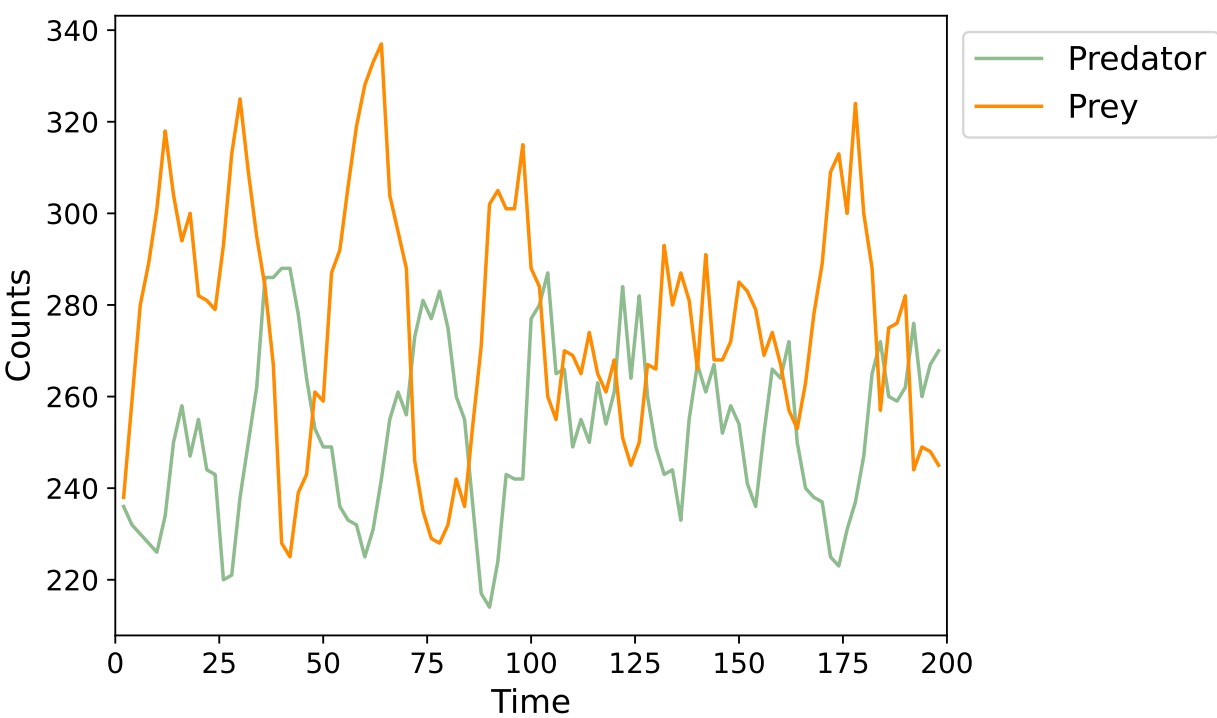

Figure 6: Predator and prey counts for $r = 0.9$. Both populations oscillate around an equilibrium in a somewhat irregular fashion.

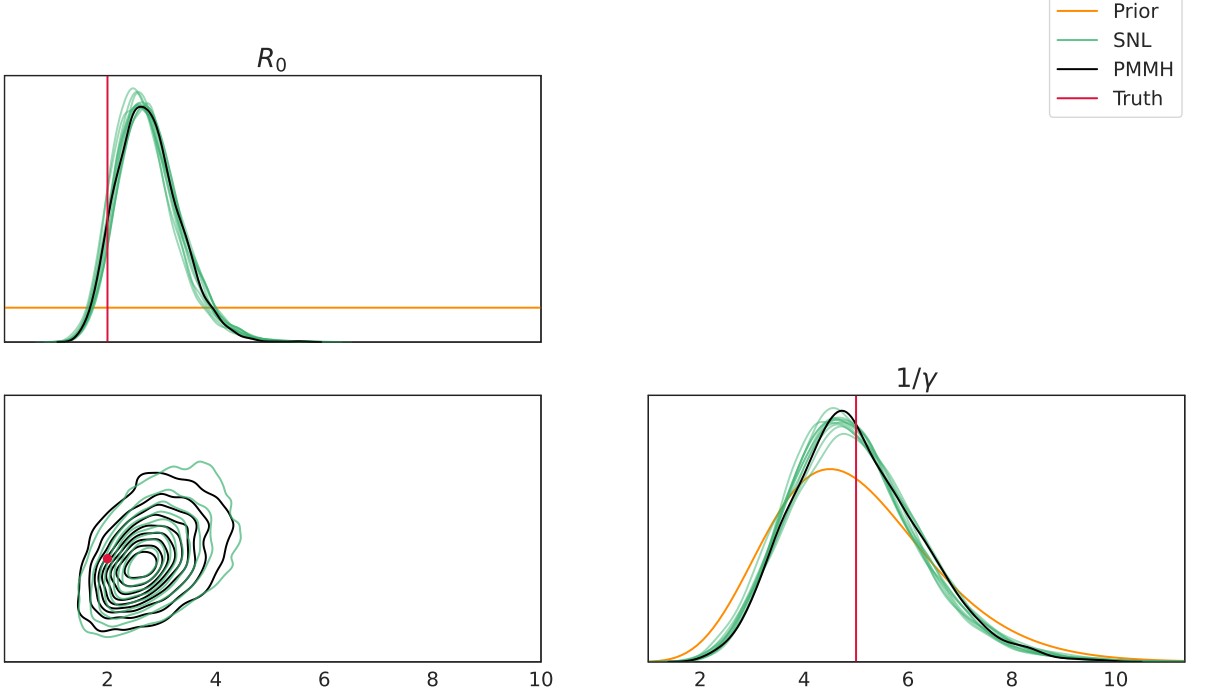

Figure 7: Posterior pairs plot for single observation from SIR model experiment.

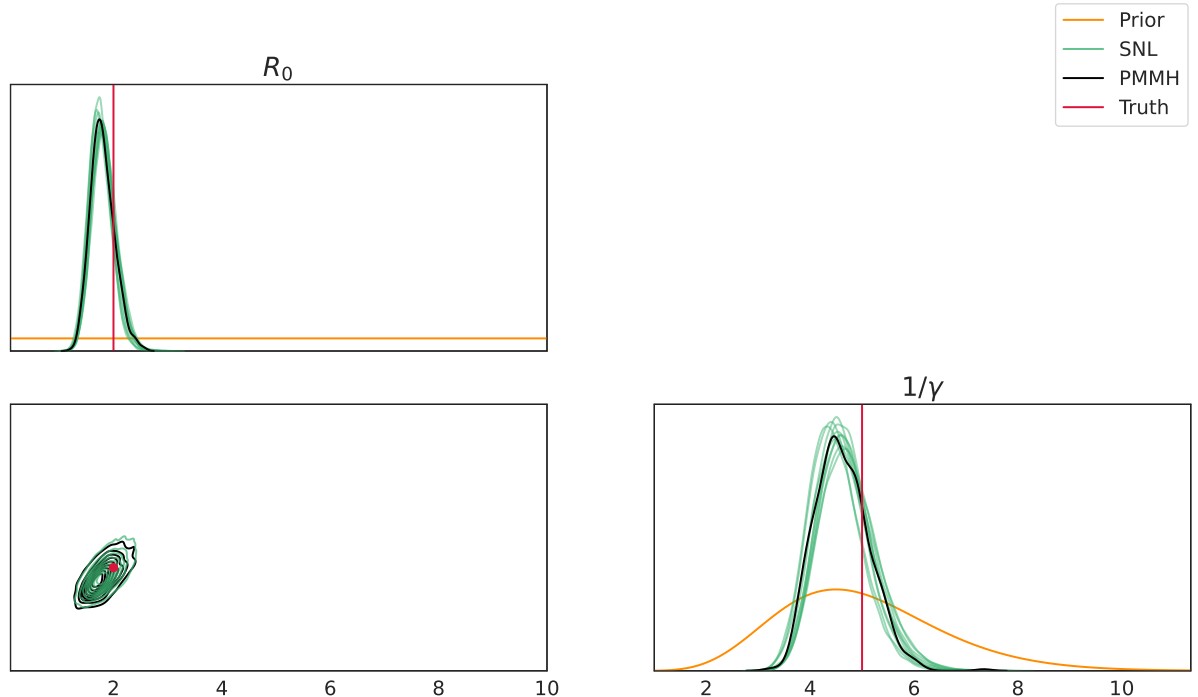

Figure 8: Posterior pairs plot for observed data from the SIR model across 100 households.

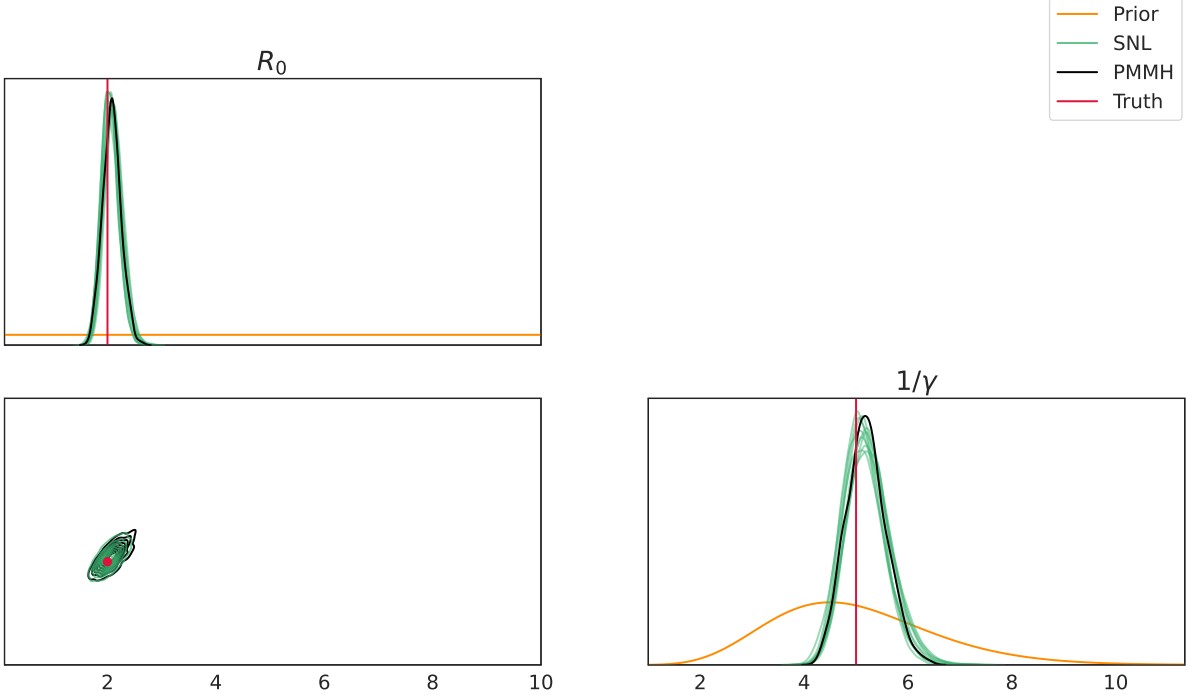

Figure 9: Posterior pairs plot for observed data from the SIR model across 200 households.

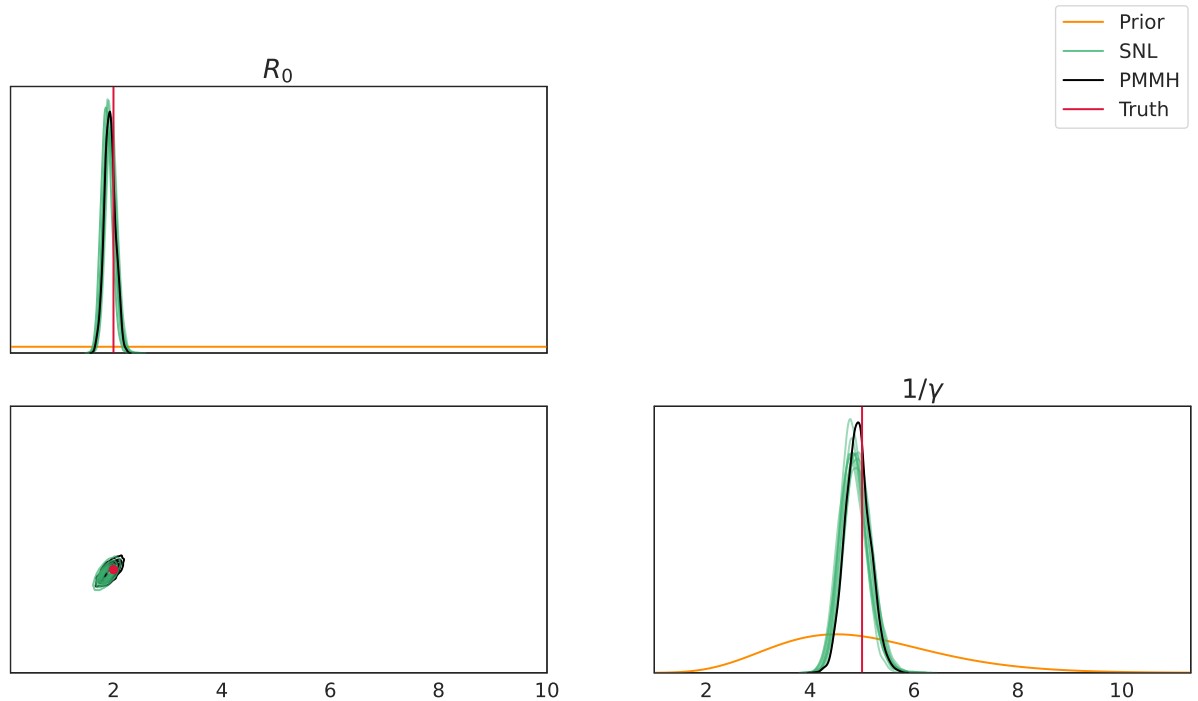

Figure 10: Posterior pairs plot for observed data from the SIR model across 500 households.

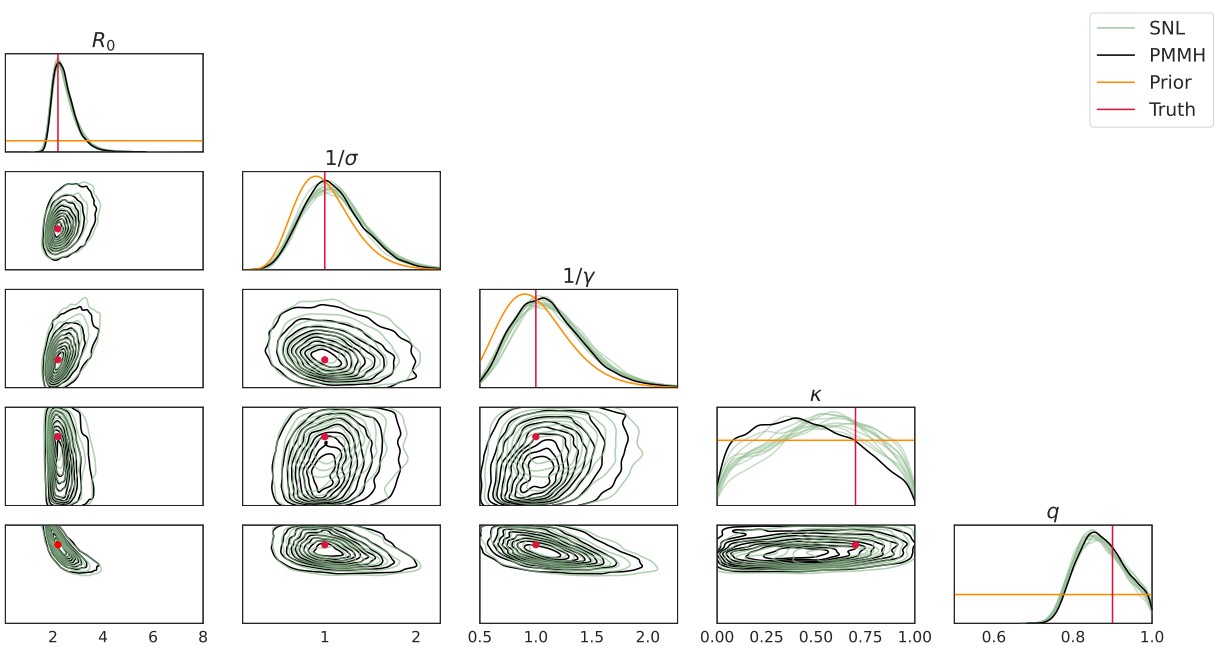

Figure 11: Posterior pairs plot for observed outbreak from SEIAR model with population size $N = 350$.

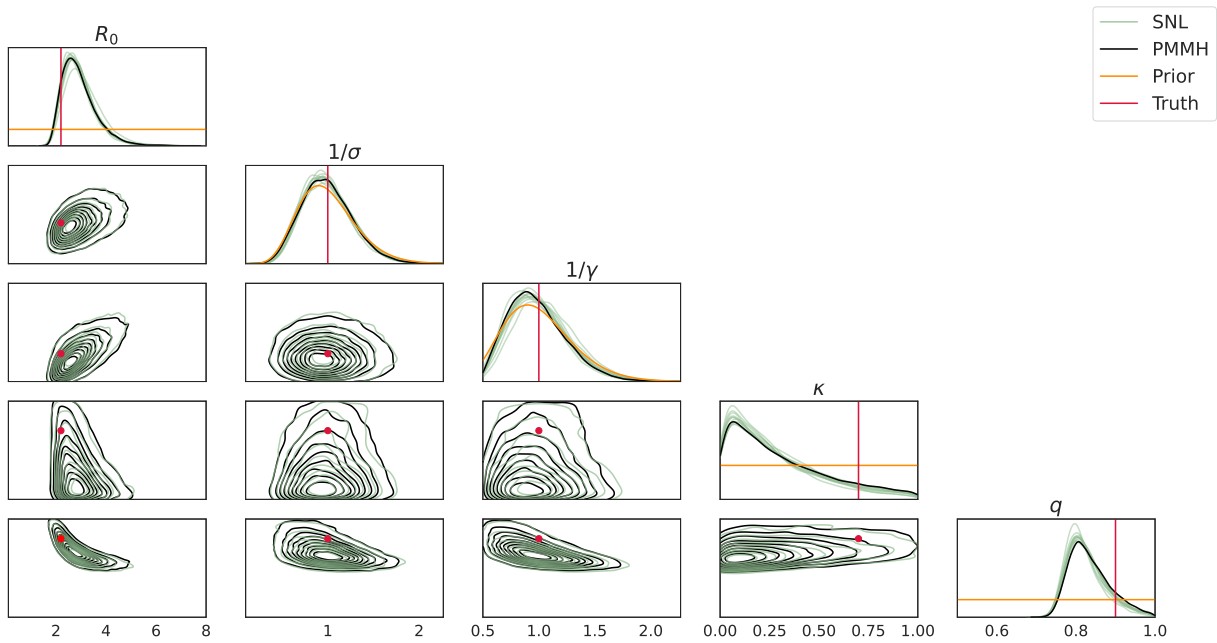

Figure 12: Posterior pairs plot for observed outbreak from SEIAR model with population size $N = 500$.

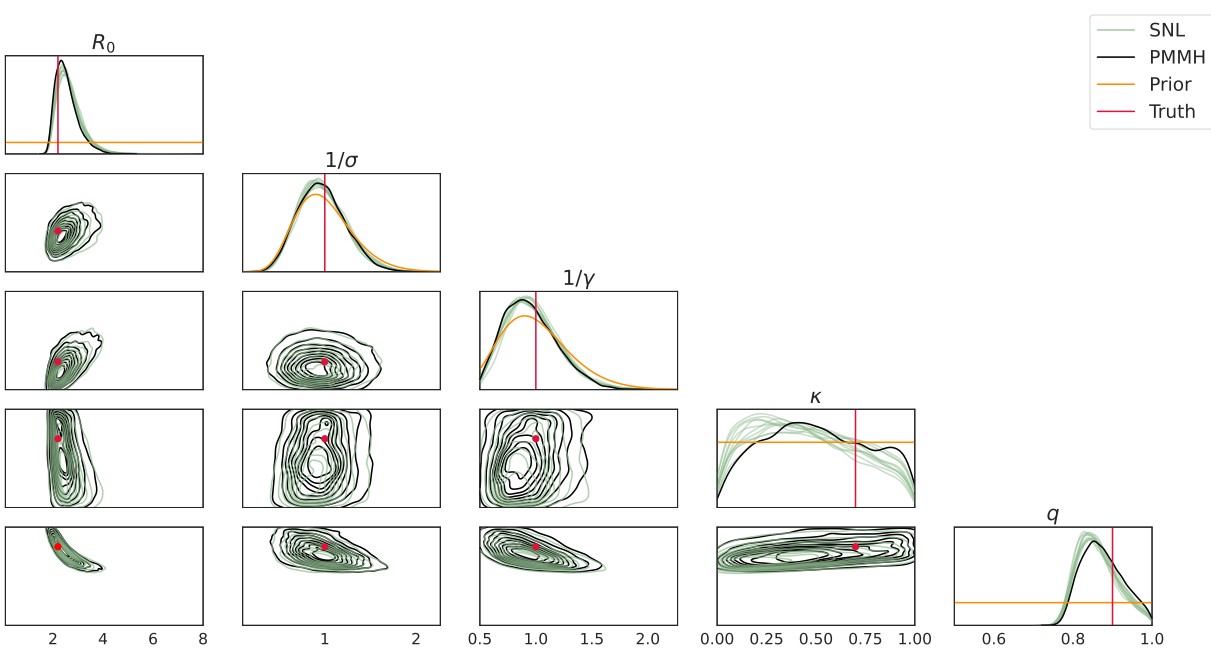

Figure 13: Posterior pairs plot for observed outbreak from SEIAR model with population size $N = 1000$.

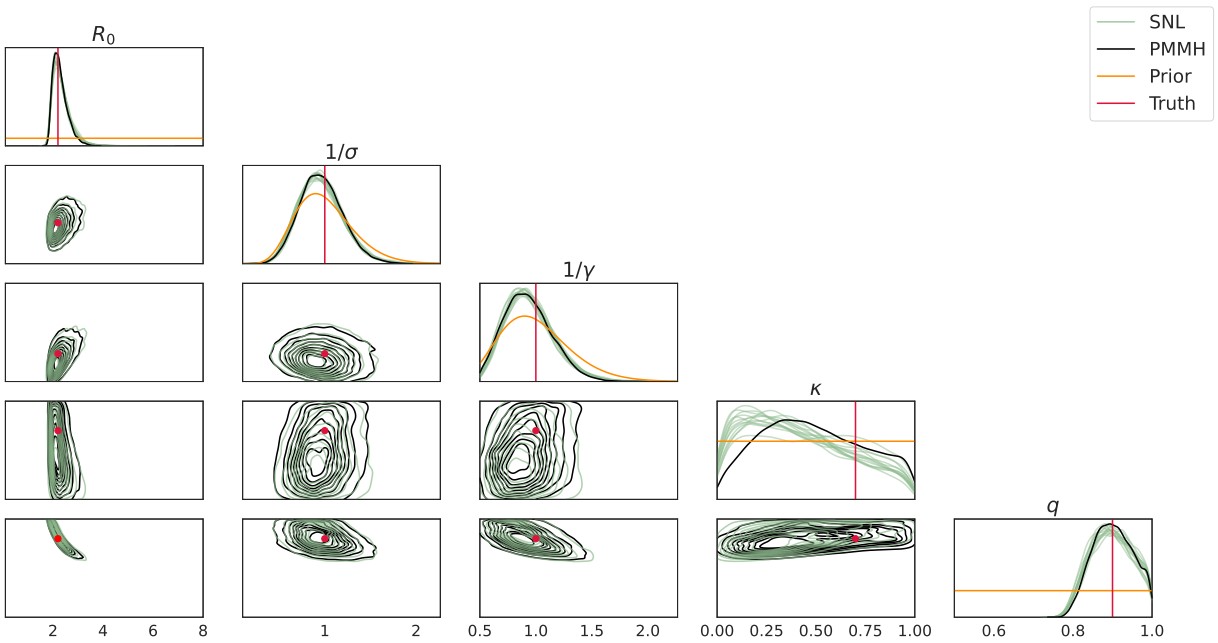

Figure 14: Posterior pairs plot for observed outbreak from SEIAR model with population size $N = 2000$.

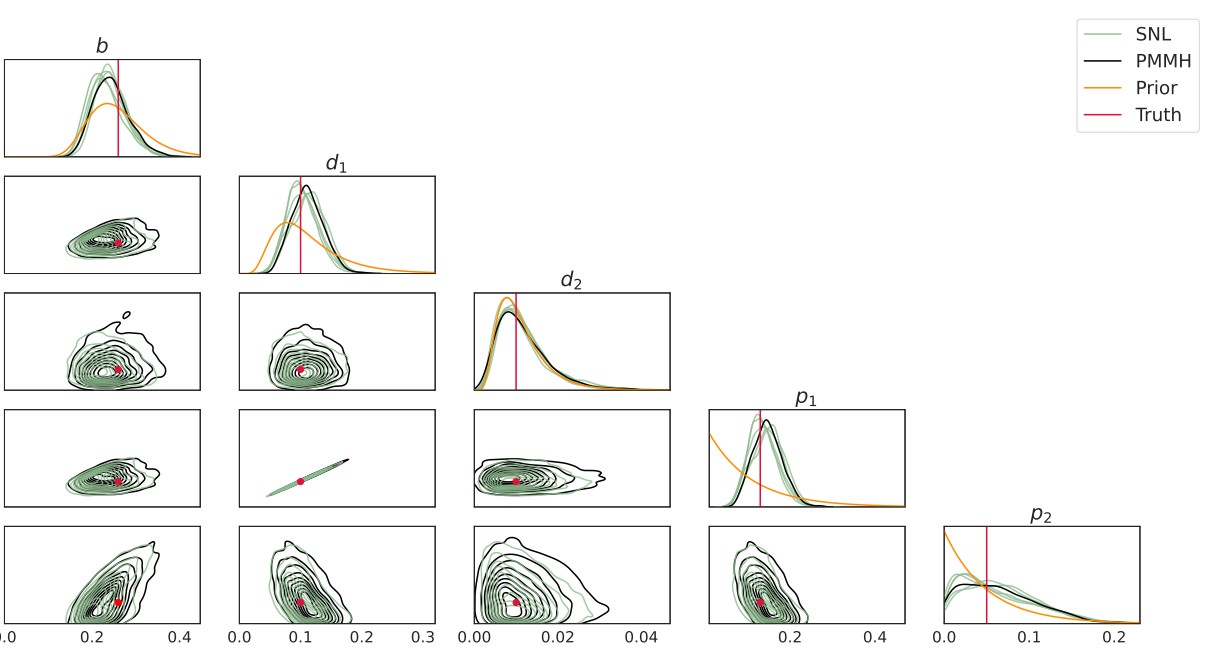

Figure 15: Posterior pairs plot for predator prey experiments with $r = 0.5$.

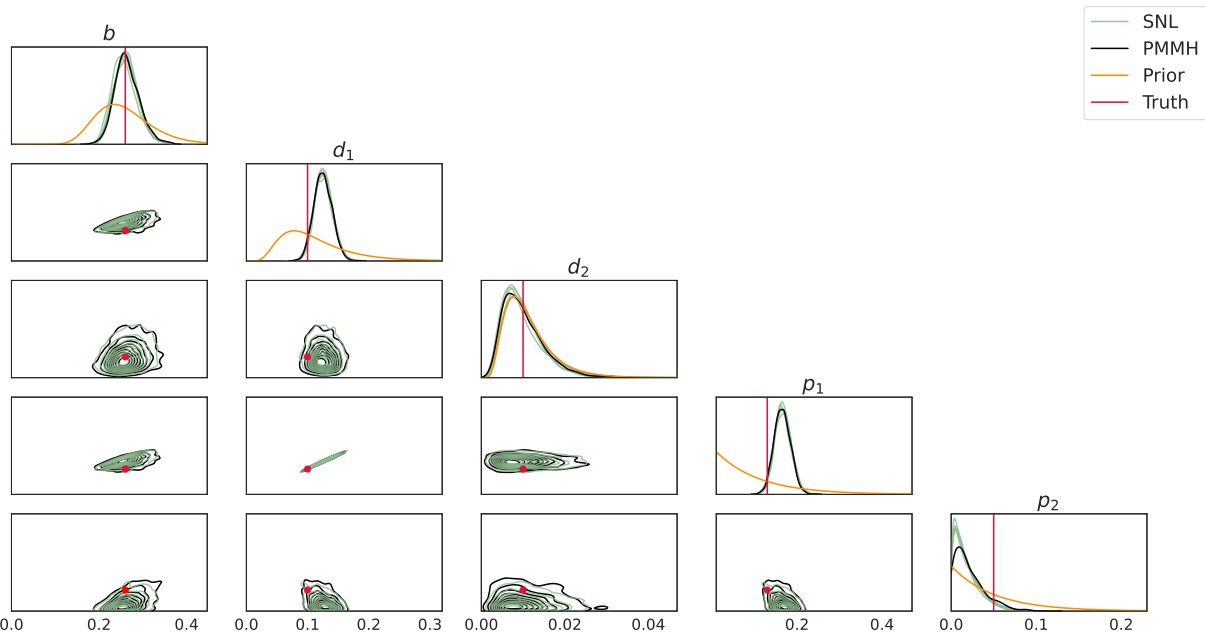

Figure 16: Posterior pairs plot for predator prey experiments with $r = 0.7$.

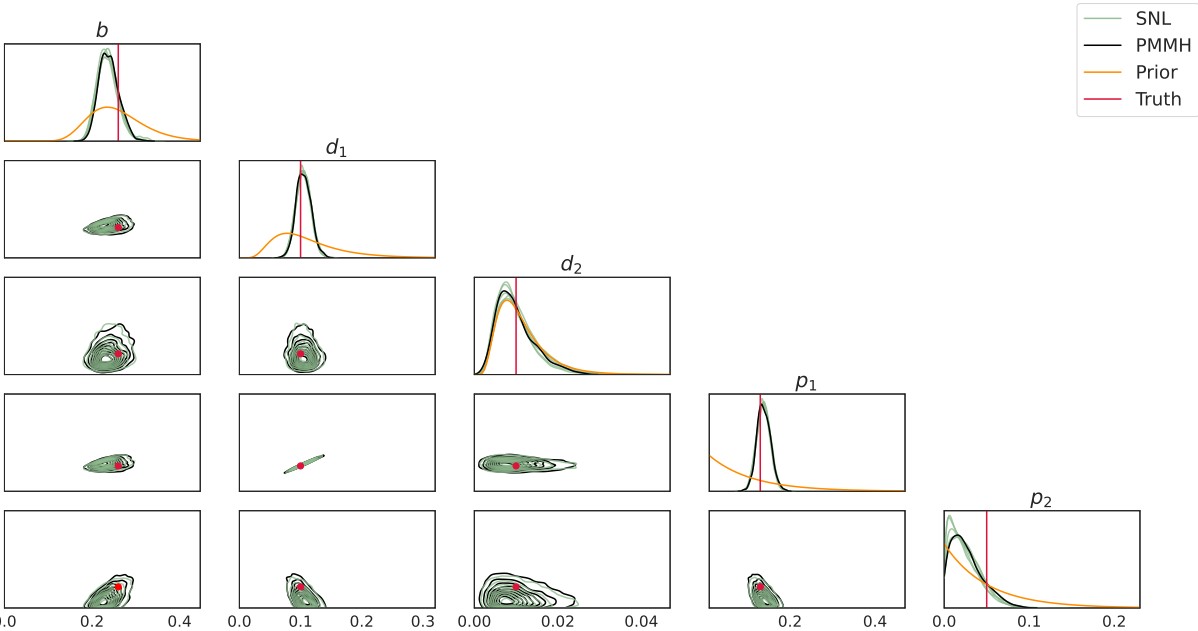

Figure 17: Posterior pairs plot for predator prey experiments with $r = 0.9$.

