# OpenReview forum: "Neural Likelihood Approximation for Integer Valued Time Series Data"
_TMLR — Accepted by TMLR_

### Review · Reviewer_7xDi · 2024-07-05

**Summary Of Contributions:**

This manuscript proposes a neural likelihood approximation method for performing Bayesian inference on integer-valued time series data, particularly focusing on stochastic processes in physical and biological sciences (with relevant constraints regarding dynamics of model population). The authors construct an autoregressive model using CNNs to approximate the likelihood of the data. This approach allows for efficient inference using unconditional simulations of the underlying model, which is simpler and computationally less expensive than current simulation-based inference methods that require conditional simulations.

**Audience:**

Yes

**Claims And Evidence:**

Yes

**Requested Changes:**

C1: I'd love to see a section on guidance on prior selection wrt. SNL, and its impact on the method's performance, possibly including a sensitivity analysis.

C2: I think it'd be great to probe the method for performance (and issues!) on significantly more complex models. I don't know what the literature in epidemiology looks like, but I presume there exist some. if not, might be good to look at earth-models from the econometrics / ecology / climate literature -- e.g. the AYS AYS model from Kittel et al., 2017, though that still somewhat simple)

C3: I think an analysis on the dynamics of feature learning wrt. feature ordering wrt. the presented problems in the manuscript would be neat to see.

### Typos

T1 (p2): becasue -> because

T2 (p7): autoregresssive -> autoregressive

**Strengths And Weaknesses:**

## Strengths

S1: the overall approach seems both clever and relatively novel approach to likelihood approximation in small population models, and the differentiability aspect makes it amenable to using plenty of exciting MCMC methods.

S2: The method demonstrates good scalability with increasing data size and system complexity, cleanly outperforming the state-of-the-art (at least wrt. PMMH and ESS, which are good baselines).

S3: The experimental section showcases a good degree of versatility, as SIR, SEIAR, and predator-prey are fairly different settings and overall the proposed method seems to be straightforwardly performant.

S4: I strongly appreciate that the focus is on raw time series data. Summary statistics are often extremely hard to get right in real-world data, and are a significant drawback of more commonly used methodologies in the literature.

S5: The neural likelihood approximation is differentiable with respect to model parameters, enabling the use of efficient MCMC methods like Hamiltonian Monte Carlo.

## Weaknesses

W1: The choice of priors seems to have a significant impact on the method's performance, and the paper doesn't provide clear guidelines on how to select appropriate priors. The authors mention that:

>In practice, we found the choice of priors to be important. For example, in the predator prey experiments, sometimes the MCMC chain would wander into regions of very low prior density, resulting in a degenerate proposal. This was easily fixed by truncating the priors to only allow parameters in a physical range.

I would have appreciated some kind of somewhat extensive sensitivity analysis wrt. this aspect. I agree that in principle it's usually not too hard to pick the right prior models, however I wonder whether there are clear failure cases that can be reported and (at least superficially) studied.

W2: There's a similar argument (as W1) to be made for the ordering of the components of $y_{i}$. Are there simple heuristics that could help alleviating the somewhat arbitrary hyperparameter optimization process here?

W3: SIR, SEIAR, and predator-prey don't really feel high-dimensional (though admittedly they seem to be extremely relevant within their own application domains, at least for SIR and SEIAR). Even though the manuscript makes some statements about scalability wrt. model parameters, it's not clear to me whether there are any traps that would arise from living in a highly-multivariate model.

W4: While the method shows good performance in reproducing posterior means, there are instances of inflated posterior variances (which the authors acknowledge), particularly for some parameters in the SEIAR model. Discussing how to more accurately capture the tail behavior of skewed posteriors would nicely enhance the manuscript. This also seems to be a problem that shows up primarily in SEIAR, so I wonder whether we'd see more of this for more complex models.

---

> ### Author Response · Authors · 2024-08-27
> **Response**
>
> Thank you very much for your review. We appreciate the positive feedback and the thoughtful comments. We have edited the manuscript to address some of your concerns, which we will elaborate on here.
>
> > W1: The choice of priors seems to have a significant impact on the method's performance, and the paper doesn't provide clear guidelines on how to select appropriate priors.
>
> and
>
> > C1: I'd love to see a section on guidance on prior selection wrt. SNL, and its impact on the method's performance, possibly including a sensitivity analysis.
>
> This comment seems to have arisen from a point we made in the discussion about the sensitivity to the priors, but on reflection we misworded this somewhat. The shape of the prior does not have a significant impact on the method’s performance, but sometimes having an unbounded prior support allows the MCMC chain to wander into regions where the surrogate is not well trained, and therefore may be wildly inaccurate. This was the issue faced in the predator-prey experiments, where the MCMC chain would get stuck in regions with extremely high birth/death rates (as these parameters had log-normal priors). In these regions of parameter space, the output of the model looks very different to the observed data we are fitting to, so presumably the surrogate likelihood has a hard time learning to resolve these differences. For example, in the SIR model if R0<1 then an outbreak will go extinct quickly.
>
> This is a problem that all SBI methods face and is not unique to the models we have tested on (see e.g. [2] from reviewer KPNs). This problem was easily fixed by truncating the support of the prior to be bounded. In practice the truncation we ended up doing for the PP model was actually quite minimal (see appendix B.4) and disallows only very unphysical parameters (extremely large birth/death rates that could not support an ecosystem in reality) that would also be ruled out easily by a practitioner. A prior predictive check is a standard way to make such choices.
>
> In addition, it was easy to detect when this occurred because the MCMC draws after early SNL rounds would report extremely small (or sometimes NaN) ESS values. The ability to quickly detect problems is a positive feature in our view.
>
> We agree that we were unclear about this in our first submission and have amended the discussion appropriately. We hope that these edits address the reviewer’s request. A sensitivity analysis should be performed in real world implementations; but for the present paper we have focused on methodology and the structure of the surrogate likelihood.

---

> ### Author Response · Authors · 2024-08-27
> **Response part 2**
>
> > C2: I think it'd be great to probe the method for performance (and issues!) on significantly more complex models. I don't know what the literature in epidemiology looks like, but I presume there exist some. if not, might be good to look at earth-models from the econometrics / ecology / climate literature -- e.g. the AYS AYS model from Kittel et al., 2017, though that still somewhat simple)
>
> and
>
> > W3: SIR, SEIAR, and predator-prey don't really feel high-dimensional (though admittedly they seem to be extremely relevant within their own application domains, at least for SIR and SEIAR). Even though the manuscript makes some statements about scalability wrt. model parameters, it's not clear to me whether there are any traps that would arise from living in a highly-multivariate model.
>
> In terms of model complexity, we would argue that the SEIAR model experiments serve this purpose within the context of this paper. This is a 5 dimensional model (compared to the 2 dimensional SIR model), and a variant of it has been used in a real world example of hepatitis A outbreaks in [1]. This model is probably at the limit of complexity that one would fit to univariate data on infection counts.
>
> Of course, it is possible to formulate arbitrarily complex epidemiological models by adding more compartments to the model, but unless additional data that is informative for transitions between these compartments is available (and it usually isn’t), then we just end up with posteriors that are very similar to the priors. Taking a Baysian approach helps with the problem of overfitting, but if the data is uninformative for most parts of the model then there is little to be gained by increasing the model complexity.
>
> Moreover, in cases where there is truly very high dimensional discrete data (e.g. from lattice models like the Ising model), there is often a lot of symmetry or large limit properties that can be exploited to find summary statistics or apply a lower-dimensional continuum approximation that would be better suited to inference than considering the high dimensional data directly.
>
> We had some difficulty tracking down the Kittle et al reference given. We believe it is this paper: https://arxiv.org/abs/1706.04542 but please correct if not. With regards to the AYS AYS model, from what we can gather it is an ODE model that produces continuous valued data, so it would not be suitable for our method that is specifically designed for discrete data. Our method could be adapted to handle such data, but this goes beyond the scope of the current paper.
>
> [1] Regan, D.G. et al. (2016), Estimating the critical immunity threshold for preventing hepatitis A outbreaks in men who have sex with men, Epidemiology and Infection, 144(7), pp. 1528-1537

---

> ### Author Response · Authors · 2024-08-27
> **Response part 3**
>
> > W2: There's a similar argument (as W1) to be made for the ordering of the components of . Are there simple heuristics that could help alleviating the somewhat arbitrary hyperparameter optimization process here?
>
> and
>
> > C1: I think an analysis on the dynamics of feature learning wrt. feature ordering wrt. the presented problems in the manuscript would be neat to see.
>
> For realistic problems, there are not actually a high number of observed dimensions, hence only a small number of orderings. This is because the time series is modeled directly and not the whole latent state. For example, epidemiological data is usually going to be a one dimensional time series of case reports. There are cases where more is available, such as predicting COVID prevalence, but one would not treat that as discrete even if it was reported as such. In similar applications to dynamical systems involving count data, we would expect that either:
>
> - The model admits a structure which makes the choice of ordering obvious, or at least significantly cuts down the choices to something manageable, e.g. systems with multiple time or spatial scales have a plausible causal relationship between the fast/fine and slow/coarse variables, and Hamiltonian systems have a plausible causal relationship between conjugate variables.
> - The likelihood can be factored in such a way that the surrogate likelihood need not treat the data as a single high-dimensional observation, e.g. epidemiological data from separate communities could be treated with a hierarchical Bayesian model, in which case the likelihood factors when conditioned on the community level parameters.
>
> So in most cases of interest, treating the order as a feature to be learned would basically come down to testing a small number of possibilities. For the PP model we considered, ordering the predators before the prey makes intuitive sense because in absence of the predators, the prey population dynamics changes from oscillatory to exponential growth, so conditioning on the predator population presumably makes it easier for the model to detect this change in dynamical behaviour.  We did try the alternative ordering, and it made little difference in this example, so we saw no point including results for the other ordering. We have updated the discussion to be clearer on these points.
>
> The problems we considered are unlike those tackled in the original MADE paper (see https://arxiv.org/abs/1502.03509) that are fitting to higher dimensional objects with no inbuilt autoregressive property. For that paper, they address the issue by randomly permuting the ordering and creating an ensemble. We did not mention this in our original submission, but will add it as a point in the discussion.
>
> > W4: While the method shows good performance in reproducing posterior means, there are instances of inflated posterior variances (which the authors acknowledge), particularly for some parameters in the SEIAR model. Discussing how to more accurately capture the tail behavior of skewed posteriors would nicely enhance the manuscript. This also seems to be a problem that shows up primarily in SEIAR, so I wonder whether we'd see more of this for more complex models.
>
>
> We appreciate the comment, and will amend the discussion to address possible approaches for better capturing the tails. For the SEIAR experiments, the SNL posterior for the main parameter of interest (R0) only has a slightly heavier tail than the PMMH posterior (see plots in appendix D). We argue that this is preferable to overconfident estimates, especially in applications where capturing the true variability is important, such as when using the resulting posteriors for decision making.
>
> We also want to point out that an approximate posterior that is slightly heavy tailed can still be useful for downstream tasks. For example, to run PMMH for the predator-prey experiments, we used the covariance of the SNL posteriors to create the Metropolis-Hastings proposal, as the parameters d1 and p1 are extremely highly correlated, which makes the pilot run to tune the MH parameters extremely costly. A posterior with overinflated variance can also be used as a good importance sampling distribution for calculating a better posterior using exact, but slower, methods (see for example: G. Papamakarios and I. Murray, Distilling Intractable Generative Models, NeurIPS workshop on Probabilistic Integration, 2015).

---

### Review · Reviewer_vfTp · 2024-07-13

**Summary Of Contributions:**

This work studies the inference of model parameters in integer-valued state space models. The authors proposed a method based on the combination of neural conditional density estimation (NCDE) and sequential neural likelihood (SNL) inference, and designed a causal convolution model for NCDE.  Simulation experiments shows that the method outperforms particle marginal Metropolis-Hastings (PMMH), a previous state-of-the-art for such models, in terms of computational efficiency.

**Audience:**

Yes

**Broader Impact Concerns:**

There is no such concern.

**Claims And Evidence:**

Yes

**Requested Changes:**

As the behavior of the method depends on the ordering of output dimensions, it may be helpful to study the sensitivity empirically.

Minor issues:

- There are several places where \cite or \citet should be \citep.  E.g. the Bickel et al (2008) on p.1.
- Appendix D: title should be capitalized consistently.
- p.5, " rounding to down to the nearest integer" -- should be "rounding down to"?

**Strengths And Weaknesses:**

Strengths:

- The paper is well-written. The method is explained clearly and appears sound.
- The method is intuitive and outperforms previous state-of-the-art in terms of scalability, and thus could be a useful addition for such applications.

Weaknesses:

- The behavior of the method depends on an ordering of observation dimensions.
- While this is not necessarily a weakness given the TMLR policies, I am somewhat uncertain about the relevance of the submission in ML venues.  The method appears obvious given the constraints of the problem, and while the model studied can certainly be important in some application fields, to me it is less clear if they are of major importance to a general ML audience, so it is not clear how they could benefit from seeing this paper in this venue.  I would note that I do not work in this specific area and am open to the opinions of other reviewers.

---

> ### Author Response · Authors · 2024-08-27
> **Response**
>
> We would like to thank you for your review, and we appreciate that you think the paper is well written. We have uploaded a revised manuscript to improve the manuscript based on the reviewer comments.
>
> For an elaboration on the changes we have made with respect to you comments, see the 3rd part of the response to reviewer 7xDi, who had a similar comment.

---

### Review · Reviewer_KPNs · 2024-08-13

**Summary Of Contributions:**

The paper presents an enhancement to simulation-based inference methods for discrete time-series data through the introduction of an autoregressive neural likelihood model. This model leverages causal convolutions to maintain the autoregressive nature of a continuous-time Markov chain and utilizes a discrete mixture of logistics to model the intractable likelihood.
The authors evaluate their approach on two epidemiological tasks and a predator-prey model involving discrete time-series observations. When compared to particle marginal Metropolis Hastings (PMMH) in a conditional inference setting, the proposed method demonstrates comparable posterior expressivity while offering superior performance in terms of the ESS/s metric.

**Audience:**

Yes

**Broader Impact Concerns:**

I don't have any broader impact concerns.

**Claims And Evidence:**

Yes

**Requested Changes:**

- I would appreciate experiments that highlight the benefits of the introduced autoregressive model/causal convoluations over a standard MLP.
- Eq(4): To my understanding, the DMoL operates with a probability mass function rather than a density function, as does the DL. In general, it should be made more clear when random variables are discrete or continuous.
- I understand the inclusion of ESS/s as a performance metric, but I believe that ESS is a particularly significant aspect of particle-based inference methods and should be reported explicitly. Additionally, it is difficult to estimate inference time directly from the ESS/s plots. Therefore, I would appreciate it if both of these quantities could be presented.
-  > The work most similar to ours is described in the original SNL paper Papamakarios et al. (2019), but a limitation of that is the use of summary statistics (as in an ABC approach) for training the model.

   While I generally agree with your statement, it’s worth noting that results of SNL on discrete time-series observations for the SIR task were reported in [1]. Despite the network architecture not being specifically tailored for discrete data, the posteriors closely matched the reference posterior data. I would be interested if you deem a comparison to the vanilla SNL be worth and if so, I would be curious how the proposed architecture compares against it.
- Please define $\mathbb{Z}$.
- It would be great if the authors could report on the simulation budget of the PMMH.

**I further list my open questions here:**
1. As someone with limited experience in CTMCs, I'm curious about your decision to use the DMoL parameterization instead of the more commonly used tokenization and embedding methods typical in transformer-based architectures or in TCN [3] in which the output would be a softmax over the state-space. Could you elaborate on this choice, especially considering the complexities involved in handling multivariate data?
2. > adding depth to the network over width since computational cost grows quadratically with width

   Shouldn’t width be only increasing in space complexity, while depth should increase the time complexity?
3. Could you clarify how the ensemble of SNL posterior is utilized? Specifically, is the ensemble treated as a mixture of posterior distributions from which samples from one of the posteriors are drawn with equal probability?
4. How can the ESS/s of PMMH and SNLE be compared if the codebases are different (Julia vs Jax)?
6. > Studying how robust our method is to model misspecification would also be important future work, as any real data is unlikely to truly follow a specific model.

   Since vanilla SNL has been demonstrated to struggle with model misspecification [2], I’m curious to know if you anticipate greater robustness with the added structure in your approach.

[1] Lueckmann, J. M., Boelts, J., Greenberg, D., Goncalves, P., & Macke, J. (2021, March). Benchmarking simulation-based inference.
[2] Kelly, R., Nott, D. J., Frazier, D. T., Warne, D., & Drovandi, C. (2024). Misspecification-robust Sequential Neural Likelihood for Simulation-based inference.
[3] Bai, S., Kolter, J. Z., & Koltun, V. (2018). An empirical evaluation of generic convolutional and recurrent networks for sequence modeling.

**Strengths And Weaknesses:**

**Strength**

- The SBI literature on time-series data with discrete observation spaces is still limited while providing important problems. Therefore,
I believe that providing a neural architecture for this specific set of tasks is a good addition to sequential neural likelihood estimation to alleviate the need to represent inter-valued time-series by summaries.

- The tasks are well suited and show that SNL with the proposed architecture approximates the PMMH posterior well. In particular, SNL does not drop performance in terms of the ESS/s when dealing with increasing number of observations and longer time-series.

**Weaknesses**

- Given that one of the paper's major contributions is the introduction of an autoregressive model for modeling the CTMC, the experiments lack an ablation study to demonstrate the advantages of this autoregressive approach compared to a standard feed-forward network.

- The extension to multivariate data seems not to be straight forward and and introduces potential biases related to the ordering of observations.

- The evaluation metrics chosen are somewhat limited. Given that the posterior results of PMMH are treated as reference data, it would be beneficial to include additional metrics [1] such as the classifier 2-sample test (C2ST) [2], which is widely used. Moreover, considering that the pair plots for the SEIAR task reveal skewed marginal densities with high pair-wise correlations, relying solely on posterior means and standard deviations may not provide the most accurate assessment.

[1] Lueckmann, J. M., Boelts, J., Greenberg, D., Goncalves, P., & Macke, J. (2021, March). Benchmarking simulation-based inference.
[2] Lopez-Paz, D., & Oquab, M. (2016). Revisiting classifier two-sample tests.

---

> ### Author Response · Authors · 2024-08-27
> **Response**
>
> Thank you for your detailed review of the manuscript. We have uploaded a revision that addresses some of the points you have raised. We will elaborate on these changes with respect to your comments and open questions here.
>
> > Given that one of the paper's major contributions is the introduction of an autoregressive model for modeling the CTMC, the experiments lack an ablation study to demonstrate the advantages of this autoregressive approach compared to a standard feed-forward network.
>
> and
>
> > I would appreciate experiments that highlight the benefits of the introduced autoregressive model/causal convoluations over a standard MLP.
>
> We appreciate that the lack of a standard network makes comparison more difficult, but it is unclear how one would incorporate the data in our experiments into a model based solely on an MLP. A feed-forward network would be unsuited to the variable length nature of the time series that were used in the epidemiological model experiments. Even for fixed length time series, a mixture-density network type architecture would not be suitable since there is no simple, computationally tractable analog of the multivariate Gaussian supported on the integers. Furthermore, the density estimation model used within the original SNL paper is based upon the model MADE, which is itself an autoregressive model, but not one designed for time series.
>
> Also note that the autoregressive model is not modelling the state of the CTMC directly, which is very large even for simple models; it is modelling the observation process which is not Markovian in general.
>
> For these reasons, it seemed best to us to compare our method to the gold standard PMMH method.
>
> > The extension to multivariate data seems not to be straight forward and introduces potential biases related to the ordering of observations.
>
> See Response part 3 to reviewer 7xDi.
>
> > The evaluation metrics chosen are somewhat limited. Given that the posterior results of PMMH are treated as reference data, it would be beneficial to include additional metrics [1] such as the classifier 2-sample test (C2ST) [2], which is widely used. Moreover, considering that the pair plots for the SEIAR task reveal skewed marginal densities with high pair-wise correlations, relying solely on posterior means and standard deviations may not provide the most accurate assessment.
>
> Fair point. We have added C2ST scores for the SEIAR experiments. Since the SIR model experiments have two dimensional posteriors, it is easy enough to visually assess the performance there, so we omitted this metric for these experiments. We omitted this metric from the predator-prey model experiments because the exact posteriors are very computationally expensive to sample from, so generating a large enough ESS to train a classifier becomes an issue.
>
> > Eq(4): To my understanding, the DMoL operates with a probability mass function rather than a density function, as does the DL. In general, it should be made more clear when random variables are discrete or continuous.
>
> Thank you for pointing this out. We have edited the manuscript to refer to probability mass functions for discrete variables.
>
> > I understand the inclusion of ESS/s as a performance metric, but I believe that ESS is a particularly significant aspect of particle-based inference methods and should be reported explicitly. Additionally, it is difficult to estimate inference time directly from the ESS/s plots. Therefore, I would appreciate it if both of these quantities could be presented.
>
> We would first like to clarify a potential point of confusion. The ESS/s reported is the ESS associated with MCMC, see e.g. Gelman et al. (2013). The ESS associated with particle filtering is a separate concept that is not really relevant to this problem beyond tuning runs of PMMH. The number of particles for the particle filter was tuned to be optimal based on a pilot run of PMMH, rather than based on the ESS of the particles. We have amended the results section to make this more clear.
>
> Reporting ESS is a sensible decision when simulations are run with a fixed computational budget, but that was not the approach in our manuscript. Instead, we ran PMMH until the ESS was at least 1,000, and the average ESS reported for SNL was always between 1,000 and 3,000 for the experiments that we ran. Our elaboration of this approach was confined to Appendix B; in response to the reviewer’s question, we have moved that elaboration to the results section and cross-reference that for each experiment. The run time may approximately be inferred by dividing 1,000 by the reported ESS/s value.

---

> ### Author Response · Authors · 2024-08-27
> **Response part 2**
>
> > While I generally agree with your statement, it’s worth noting that results of SNL on discrete time-series observations for the SIR task were reported in [1]. Despite the network architecture not being specifically tailored for discrete data, the posteriors closely matched the reference posterior data. I would be interested if you deem a comparison to the vanilla SNL be worth and if so, I would be curious how the proposed architecture compares against it.
>
> For the reasons stated in the first comment, it would not be possible to apply vanilla SNL without using summary statistics. The SIR task in the reference cited is different from ours in a few key ways:
>
> 1. They use a continuous-state ODE model which is only valid for large population sizes. We consider small infected population sizes and a discrete model; the ODE model would be a poor approximation for our situation.
> 2. Their observations are conditionally independent given the model parameters, as the parameters uniquely determine the trajectories, and the observations are conditionally independent given the trajectories (not the case with the stochastic model). This would make using an autoregressive model redundant.
> 3. They use a fixed length time series. Again, this is not possible with the stochastic model because outbreaks can fade out in the first couple of time steps, or they can continue growing for a lengthy period.
> 4. They do not incorporate final size data. The final size is informative for R0 and its inclusion allows us to prevent the model from trying to learn from data after the outbreak has ended (where you just have a constant sequence).
>
> So, it is unfortunately not possible to compare the two sets of experiments, as the data is quite different and the stochastic model is more complex to deal with than the ODE model.
>
> > It would be great if the authors could report on the simulation budget of the PMMH.
>
> We thank the reviewer for this comment. We should have made it more clear that we ran PMMH for as long as required to generate the target ESS of 1000 per parameter, rather than using a fixed simulation budget. We have addressed this by including a discussion on simulation budget in section 4 (pg. 9).
>
> > As someone with limited experience in CTMCs, I'm curious about your decision to use the DMoL parameterization instead of the more commonly used tokenization and embedding methods typical in transformer-based architectures or in TCN [3] in which the output would be a softmax over the state-space. Could you elaborate on this choice, especially considering the complexities involved in handling multivariate data?
>
> We used a DMoL because it reduces the number of parameters that the surrogate has to learn, and exploits the fact that the conditionals will ‘look like’ a discretisation of a continuous distribution (i.e. most of the mass will be concentrated around one or two modes). For examples where the state space of the Markov chain is large (or possibly infinite), tokenization does not seem like a natural choice. For the household experiments where the population is very small, tokenization and using a multinomial may work quite well (and possibly even be more efficient than the DMoL). We have added this point in a sentence at the end of section 5.
>
> The complexities involving multivariate data have to do with the fact that discretising multidimensional distributions with correlations is hard because you must calculate an infinite sum without a closed form to normalise it. A softmax over the observation space would work in theory, but the output dimension would then grow exponentially (rather than linearly) with the dimension. For example, in the predator prey experiments we used a carrying capacity of 800 for both populations, so the space of observations contains 800*800=640,000 distinct states.
>
> > Shouldn’t width be only increasing in space complexity, while depth should increase the time complexity?
>
> Width was a poor choice of words here, as what we were referring to was the number of channels in the CNN. We have adjusted the manuscript to make this more clear.
>
> > Could you clarify how the ensemble of SNL posterior is utilized? Specifically, is the ensemble treated as a mixture of posterior distributions from which samples from one of the posteriors are drawn with equal probability?
>
> Correct. We have edited the text to make this more clear.

---

> ### Author Response · Authors · 2024-08-27
> **Response part 3**
>
> > How can the ESS/s of PMMH and SNLE be compared if the codebases are different (Julia vs Jax)?
>
> The different nature of these tasks made it preferable to use different languages for the SNL and PMMH implementations, and we attempted to use the optimal implementation of each method. The PMMH particle filter used was originally written in C and the Julia implementation is close in performance, and it would not be easy to implement using the tools available in Jax. We used Jax for SNL because it allowed for a compiled version of NUTS to be used with the neural network through the numpyro library. As the particle filter used for the epidemiological experiments is not suited to the tools available in Jax, and our SNL implementation relies on Jax + numpyro, we cannot bring the algorithms under one language.
>
> > Since vanilla SNL has been demonstrated to struggle with model misspecification [2], I’m curious to know if you anticipate greater robustness with the added structure in your approach.
>
> We believe that greater robustness is an unlikely outcome. Even though we don’t use summary statistics, the data is still partially observed, which would presumably make it more likely to fit a misspecified model (even in an exact approach). The surrogate can also still seemingly learn to be overconfident in regions of parameter space far away from most of the prior mass (see first point in reply to reviewer 1). This suggests that the surrogate is still capable of being overconfident under model misspecification. We have added a comment in the manuscript discussion that more robust training methods are an important topic of current research and added a reference to [2].

---

### Author Response · Authors · 2024-08-27
**General response**

We would like to thank the reviewers for dedicating their time to thoroughly review our manuscript and provide us with useful feedback.

We have uploaded a new version of the manuscript to address the given feedback. All minor corrections and typos highlighted by the reviewers have been fixed, and in addition the new manuscript includes:

- An enhanced discussion section with new discussion points
- Rephrasing of some of the existing discussion to clear up issues brought up by reviewers around choice of feature orderings and priors.
- Clarification of some details in the results section that were missing or hidden in the appendix beforehand.
- Added C2ST score as performance metric for SEIAR model experiments.

---

### Decision · Action_Editor_J45h · 2024-10-07

**Recommendation:** Accept as is

**Comment:**

Based on the positive recommendation of all three reviewers, i suggest acceptance.

**Audience:**

The manuscript is clearly interesting for the Simulation based Inference community.

**Claims And Evidence:**

The claims made in the manuscript i supported by convincing evidence.